

# PalVol v1: A proxy-based semi-stochastic ensemble reconstruction of volcanic stratospheric sulfur injection for the last glacial cycle (130,000 – 50 BP)

Julie Christin Schindlbeck-Belo[1]*, Matthew Toohey[2], Marion Jegen[1], Steffen Kutterolf[1], Kira Rehfeld[3]

[1]GEOMAR Helmholtz Centre for Ocean Research Kiel, Wischhofstr.1-3, 24148 Kiel, Germany
[2]Institute of Space and Atmospheric Studies, Department of Physics & Engineering Physics, University of Saskatchewan, S7N 5A2 Saskatoon, Canada
[3]Department of Geosciences and Department of Physics, Geo- und Umweltforschungszentrum, Schnarrenbergstr. 94/96, Tübingen University, 72076 Tübingen, Germany

*Correspondence to*: Julie C. Belo (jbelo@geomar.de)

**Abstract.**

Perturbations in stratospheric aerosol due to explosive volcanic eruptions are a primary contributor to natural climate variability. Observations of stratospheric aerosol are available for the past decades, and information from ice cores has been used to derive estimates of stratospheric sulfur injections and aerosol optical depth over the Holocene (approximately 10,000 BP to present) and into the last glacial period, extending back to 60,000 BP. Tephra records of past volcanism, compared to ice cores, are less complete, but extend much further into the past. To support model studies of the potential impacts of explosive volcanism on climate variability over across timescales, we present here an ensemble reconstruction of volcanic stratospheric sulfur injection (VSSI) over the last 130,000 years that is based primarily on terrestrial and marine tephra records. VSSI values are computed as a simple function of eruption magnitude, based on VSSI estimates from ice cores and satellite observations for identified eruptions. To correct for the incompleteness of the tephra record we include stochastically generated synthetic eruptions, assuming a constant background eruption frequency from the ice core Holocene record. While the reconstruction often differs from ice core estimates for specific eruptions due to uncertainties in the data used and reconstruction method, it shows good agreement with an ice core based VSSI reconstruction in terms of millennial-scale cumulative VSSI variations over the Holocene. The PalVol reconstruction provides a new basis to test the contributions of forced vs. unforced natural variability to the spectrum of climate, and the mechanisms leading to abrupt transitions in the palaeoclimate record with low-to-high complexity climate models. The PalVol volcanic forcing reconstruction is available at https://doi.org/10.26050/WDCC/PalVolv1 (Toohey, Schindlbeck-Belo, 2023).



# 1 Introduction

Explosive volcanic eruptions transfer massive quantities of material from the solid Earth into the atmosphere. Eruptive plumes contain large amounts of solid material, as well as gaseous compounds including water vapor, carbon dioxide and sulfur-containing species (mostly $SO_2$), often being combined into aerosols. The solid volcanic fragments are fragmented magma that is ejected by an eruption and are called tephra (mostly ash: ø< 2mm, and lapilli: ø2 to 64 mm). They generally fall back to the surface of the Earth rather quickly, producing a tephra layer with decreasing thickness and grain size with increasing distance to the volcano. Such deposits persist as a record of past volcanic eruptions that can be seen in outcrops, or in sediment cores extracted from marine and lacustrine environments (e.g., Kutterolf et al., 2016; Schindlbeck et al., 2016).

Gaseous emissions from eruptions can persist in the atmosphere much longer than solid emissions. While the emission of $H_2O$ and $CO_2$ from a single eruption is generally insignificant compared to the atmosphere burden of these species, volcanic emissions of sulfur-containing species can produce significant increases in atmospheric sulfur. Under atmospheric conditions, sulfur-containing species produce sulfate aerosol particles, small droplets of sulfate ($SO_4$) and water in liquid form that are dispersed in a gaseous matrix. Sulfur emissions to the troposphere (from volcanic eruptions or anthropogenic activities) produce sulfate aerosols that have an atmospheric lifetime of days to weeks, as they grow in size and are eventually "rained out". Sulfate aerosols in the cold and dry lower stratosphere do not generally grow as large as those in the troposphere, and as a result persist in the stratosphere for months to years, over which time they are transported around the globe. These sulfate aerosols have important impacts on atmospheric radiative transfer, by scattering solar radiation and absorbing longwave radiation, with the net effect of a decrease in downwelling net radiation at the Earth's surface, which leads to a cooling of surface temperatures (Robock, 2000).

Post-volcanic large-scale (global or hemispheric) cooling has been observed after recent eruptions (e.g., El Chichón in 1982, Mt. Pinatubo in 1991), and is apparent in millennial scale climate reconstructions, for example the 1815 Tambora eruption in Indonesia (e.g., Rampino and Self, 1982, 1993). It has been shown that negative radiative forcing from volcanic aerosol perturbations is the primary driver of natural climate variability over the past thousand to two thousand years (Sigl et al., 2015; Schurer et al., 2013). Additionally, Kobashi et al. (2017) showed that also during the Holocene, volcanic eruptions played an important role in centennial to millennial temperature variability on Greenland. Representing the intermittent natural forcing in climate model experiments for the late Holocene and the Glacial improves modeled variability on timescales from decades to centuries (Ellerhoff et al., 2022).

The role of volcanic eruptions in longer-term climate variability has long been speculated but remains poorly understood. Strong volcanic eruptions have been linked to multi-decadal periods of cooler than usual surface temperatures, for example during the Little Ice Age (Owens et al., 2017; Miller et al., 2012; Zanchettin et al., 2013; Schurer et al., 2014; Timmreck et al., 2021) and the so-called "Late Antiquity Little Ice Age (LALIA, Büntgen et al., 2020, Toohey et al., 2016). The manifestation of volcanic radiative forcing as multi-decadal temperature anomalies has been suggested to be related to the thermal inertia of the Earth's oceans, which dampen the initial temperature response, but also prolong it through the accumulation of negative



energy input in the deep ocean (Gupta et al., 2018). Indeed, ocean sea surface temperature changes over the 801-1800 CE period are well explained by the time series of volcanic eruptions (McGregor et al., 2015). There are climate modeling studies that suggest that strong volcanic radiative forcing can perturb ocean circulation modes (Swingedouw et al., 2017), which may produce long term perturbations to surface climate (Miller et al.,2014; Schleussner and Feulner, 2013; Schleussner et al., 2015;

Otterå et al., 2010, Zhong et al., 2011). Indeed, at the global scale, clusters of eruptions have been linked to global mean temperature variations over the Common Era (Pages2k-Consortium, 2019). But other work has suggested a potential match in the timings of large eruptions or clusters of eruptions with the sudden transitions in climate between stadials and interstadials (Baldini et al., 2015; Bay et al., 2006; Lohmann and Svensson, 2022), although the robustness of this connection is limited by uncertainties in eruption magnitudes and timings.

Ice core sulfur (Huybers and Langmuir, 2009) and tephra chronologies (e.g., Praetorius et al., 2016; Sigvaldason et al., 1992) both attest to a marked increase in eruption frequencies during and after the last glaciation, especially in the northern hemisphere (NH) mid-to-high latitudes. On longer timescales, variations in eruption frequency from deep marine cores suggest periodicities on Milankovic time scales (Kutterolf et al., 2013, 2019; Schindlbeck et al., 2018a), suggesting a connection between the climate changes brought about by variations in orbital parameters and volcanic eruption frequencies. A leading

theory is that mass transfer between ice sheets and the ocean due to changing temperatures leads to changes in the pressure on the Earth's crust which can modulate crustal stress fields and enhance the possibility of associated eruptive events by providing pathways for the magma to rise (Kutterolf et al., 2013).

Recent ice core-based reconstructions of volcanic sulfur emissions confirm the increase in explosive eruptive frequency after the last deglaciation, and its likely impact on stratospheric aerosol levels (Sigl et al., 2022). Ice core data presented by Lin et

al. (2022) extends into the last glacial period, and suggests that during glacial conditions, the frequency of large eruptions with significant VSSI was similar to that after the deglaciation.

The emissions of subaerial eruptions are eventually deposited to the Earth's surface and in some cases preserved providing us with records of past volcanic activity. Tephra layers from past eruptions are often discernable in terrestrial outcrops, but the records are often incomplete since younger deposits overlay the stratigraphy or the deposits are already heavily weathered,

eroded, or covered by vegetation. Ash and tephra, either transported by fallout or density flow processes, are well-preserved in the marine sediments, since wide areas of the seafloor are relatively unaffected by erosion or bioturbation (Freundt et al., 2021). This makes marine sediments outstanding archives for previous eruptions. By piston or gravity coring (~up to 20 m depth) or drilling (several hundreds of meters) the marine sediment records can be recovered and the marine ash layers provide a stratigraphically controlled record. While drilling is time-consuming and expensive, and the lateral coverage thus limited to

a few sites, multiple shorter piston or gravity cores can be taken in a certain region providing good coverage, which enables detection of lateral stratigraphic changes or local erosion. Due to their length, however, they are often limited to the last two glacial cycles. In general, marine tephra records from sediment cores can cover several millions of years depending on the used coring/drilling technique. There are geographical gaps, or regions with sparse coverage of cores and by far not all cores have been studied for their tephra inventory in detail.

Ice cores (Fig. 1) provide a good archive for past volcanic eruptions, as the volcanic sulfate aerosols and in some cases ash
particles are deposited to the surface and incorporated into the glacial ice. Chemical analyses of ice cores therefore provide
time series, which are especially valuable when the dating of the ice cores is of high quality (Hammer, 1977; Gao et al., 2008;
Sigl et al., 2015). Past volcanic eruptions are evidenced by strong increases in sulfur/sulfate content of the ice, or in the
electrical conductivity of the ice, as the analysis traverses the ice core and thus reaches backwards in time. By synchronizing

multiple ice cores from Greenland and Antarctica, recent efforts have produced estimates of the volcanic stratospheric sulfur
injection (VSSI) from volcanic eruptions covering the past 2,500 years (Toohey and Sigl, 2017), the Holocene (Cole-Dai et
al., 2021; Sigl et al., 2015), and the late glacial period and deglaciation (Lin et al., 2022).

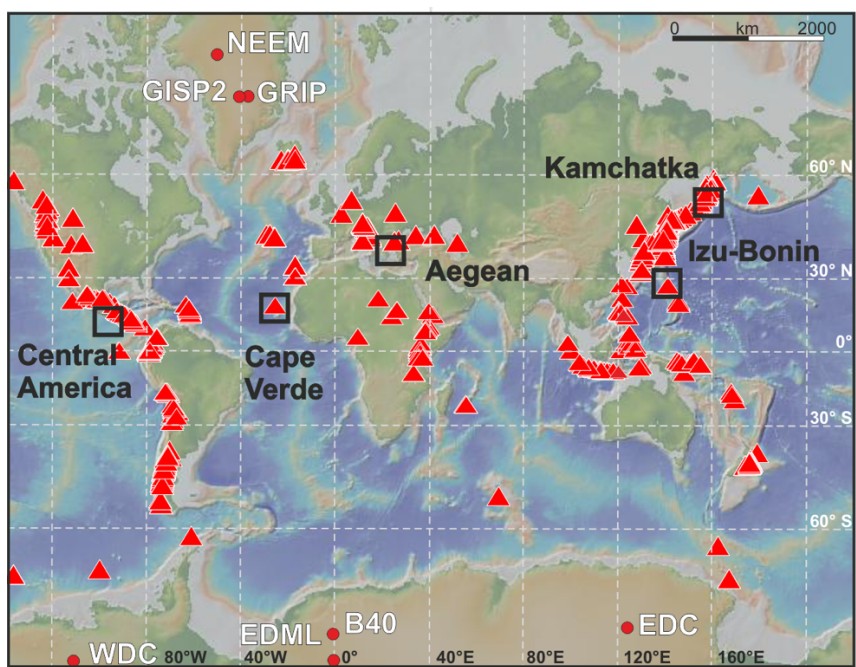

**Figure 1: Global overview map created using GeoMapApp (http://www.geomapapp.org; GMRT-Global Multi-Resolution
Topography) (Ryan et al., ,2009). Red triangles mark global distribution of volcanoes from the LaMEVE database (Crosweller et
al., 2012). Black squares mark regions with marine sediment cores that have been used in this publication. Red circles show positions
of ice cores in Greenland and Antarctica used by Cole-Dai et al. (2021) and Sigl et al. (2015, 2022 and references therein).**

Due to limitations with ice cores–absolute length, thinning, and synchronization, there is a limit to the length of time ice cores
can be used to reconstruct past volcanism. Tephra records from sediment cores, on the other hand, extend much further back
in time. However, they have their own limitations: incompleteness, dating uncertainty, not as direct a measure of the
stratospheric sulfate aerosol load as ice core sulfate records are. There is a strong temporal trend in the dataset, which is
described by a decreasing number of detected events going back in time (Fig. 2). Brown et al. (2014) for example emphasize

in the LaMEVE database, which covers the last 1.8 Ma, about 40% of the detected eruptions included occurred during the

Holocene (the past 11 ka), and they conclude that the decrease going back in time is mainly due to under-recording of eruptions. The time trend in underreporting is found to depend on the magnitude of eruption, with the frequency of smaller eruptions (M=4) falling off much faster than that for larger eruptions (M>6) (Fig. 2). Despite these limitations, far reaching tephra records do provide valuable information about specific events (e.g., Pinatubo, Toba) and changes in eruption frequency with

time. For example, analysis of tephra records long enough to cover several glacial cycles, show that eruption frequencies vary on periods representative of Milankovitch cycles, supporting the claims that the Earth's climate influences eruption frequencies on long time scales through changes in sea level and associated crustal stresses (Paterne et al., 1990; Rampino et al., 1979; Kutterolf et al., 2013, 2019; Schindlbeck et al., 2018a).

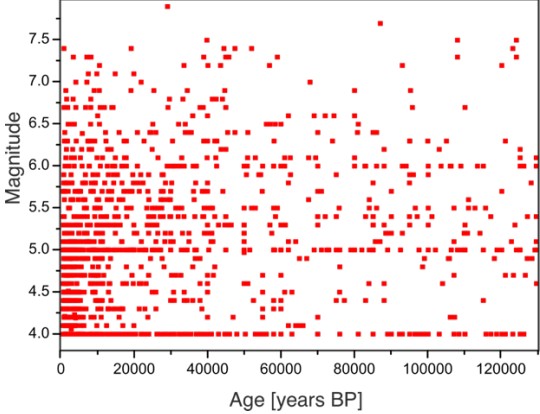

**Figure 2: Scatter plot showing the distribution of eruptions by magnitude over the last 130,000 years. Data comprises all eruptions from the LaMEVE database as well as additional eruptions from marine sediment cores as described in Section 2.1.**

Here we present a new time series of volcanic stratospheric sulfur injection called "PalVol" (data published, add reference) covering the last glacial cycle that is based on terrestrial and marine tephra records, and includes stochastically generated "synthetic" eruptions to correct for the incompleteness of tephra records, in particular regarding small to medium eruptions.

We provide an ensemble of time series, with each realization providing different timings of events found in the tephra record according to the uncertainties in the dating of the events, as well as different timings of the synthetic events. Our aim is not to provide an accurate reconstruction of the actual timing and magnitude of eruptions over the last glacial cycle. This is potentially even impossible. Rather, our aim is to provide a plausible set of such time series, each of which might approximate a possible true history given the information available and some basic assumptions. While we do not guarantee the accuracy of the timing

and magnitude of specific eruptions, we do aim to produce a time series which represents our best estimate of the stochastic forcing provided by volcanic eruptions as well as some accuracy in terms of millennial-scale variability in volcanic forcing.

The paper is organized as follows: in Section 2, the data and methods used to produce the ensemble VSSI time series product are introduced. In Section 3 the ensemble VSSI product is compared to existing ice core-based reconstructions. Discussion and conclusions are included in Section 4.





## 2 Data and methods

### 2.1 Tephra Data

The majority of tephra data used in this study is extracted from the LaMEVE database (Crosweller et al., 2012; LaMEVE Version 3), a global compilation of large magnitude eruptions with VEI 4 (Volcanic Explosivity Index), and/or magnitude 4 that are known from terrestrial deposits and outcrops. The LaMEVE database is publicly available and summarizes information regarding (1) erupted mass/volume and therefore magnitude/VEI, (2) eruption dates as well as the applied dating techniques and uncertainties, (3) the source volcano, (4) eruption parameters (e.g., column height), and (5) rock types. We focused on the time interval from 130,700 years BP to 2014 AD (comprising 1,501 events).

The eruption data taken from LaMEVE is complemented and corrected by records from a suite of marine cores and recent studies (Fig. 1; appendices tables A1, A2), including cores and samples from Central America, offshore the Izu-Bonin volcanic arc, offshore the Cape Verdes, offshore the Kamchatka peninsula, and from the Hellenic Arc. Eruptive volumes and magnitudes for eruptions with no published values have been calculated applying a single isopach approach following the methods described in Schindlbeck et al. (2018b).

The tephra record is characterized by increasing incompleteness back in time (Papale, 2018; Kiyosugi et al., 2015; Brown et al., 2014) (Fig. 2). The reasons for missing eruptions in terrestrial tephra records are mostly erosion and alteration, or burial by younger deposits (e.g., Lavigne et al., 2004; Pollard et al., 2003). However, generally it is thought that the incompleteness is more apparent for smaller magnitude eruptions (e.g., M=4) compared to larger magnitudes (e.g., M>=5) (Brown et al., 2014) since preservation in marine sediments and on land is strongly increasing with the thicknesses of deposits (Wetzel, 2009; Freundt et al., 2021) (Fig. 2). In the marine environment, increasing time for bioturbation and diagenesis (time in which bio-organisms and chemical interaction with formation waters can actively modify the ash layers physically and chemically) as well as the depositional environment play a role for incompleteness in the records (Hopkins et al., 2020; Freundt et al., 2021). For long records also the plate motions (except of ocean island volcanoes) might be responsible for a decreasing amount of recorded eruptions in the past since most of the eruptive records are coupled with convergent margins and associated arc volcanism (e.g., Schindlbeck et al., 2015). The deeper and older the ash layers in the marine sediments are, the greater the distance between the volcanic source and deposition location at the time of the eruption due to plate motion (e.g., Schindlbeck et al. 2015). This mechanism could and therefore reduce the possibility that smaller eruptions are recorded in the marine sediment archive from convergent margins.

Marine or lacustrine ash layers as well as terrestrial deposits can be either directly or indirectly dated. The techniques comprise radiometric techniques (e.g., radiocarbon ($^{14}$C), $^{40}$Ar/$^{39}$Ar mineral dating, zircon dating), orbital tuning of oxygen isotope curves, sedimentation rates, (Drexler et al., 1980; Kutterolf et al., 2008), which lead to age uncertainties of approximately 1-10% of the estimated age (Fig. 3). The most precise dates are obtained for tephra that can be associated with historical observations of an eruption or with ice core or tree ring signals.

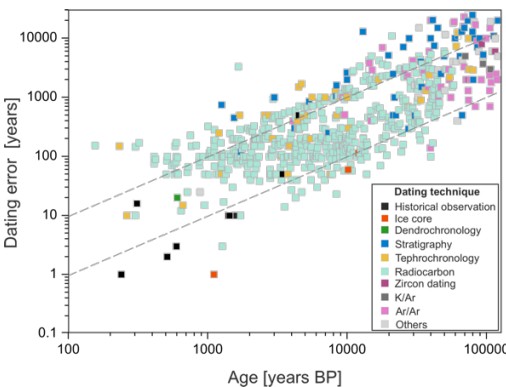

**Figure 3: Dating uncertainties over time for eruptions included in the LaMEVE data base over the X-Y period, color-coded according to the different dating techniques as provided by Crosweller et al. (2012), Cisneros et al. (2021a,b). Gray dashed lines indicate the 1% and 10% error range.**

Magnitude (M) is the preferred measure of eruption size used in the LaMEVE database (Crosweller et al., 2012), and is the quantity used here to estimate VSSI (Walker, 1980; Newhall and Self, 1982). Magnitude is a function of erupted mass, which is typically derived from estimates of erupted volumes, classically calculated by drawing isopach maps (areas of equal thickness of the ash layer; e.g., Bonadonna and Houghton, 2005; Fierstein and Nathenson, 1992; Pyle, 1995a) and integrating over area. For eruptions used in this study which are not included in LaMEVE, magnitudes are taken from the respective publications. For the volcanic eruptions described in Derkachev et al. (2020) no volume estimates were available and we therefore calculated minimum volumes applying a single isopach approach (Schindlbeck et al., 2018b) based on the given thickness data. The estimated uncertainty in mass estimates (and thus magnitude) are of the order of 10 to 20% (e.g., Kutterolf et al., 2021b; Klawonn et al., 2014), and depends on the sample size (available outcrops on land and core density in the sea), and uncertainties in rock densities used for converting volumes into masses.

## 2.2 Ice core-based VSSI

VSSI time series derived from polar ice cores are used in this work, both as a basis for the PalVol VSSI reconstruction methods (see Section 2.3, 2.4) and to validate the new reconstruction through statistical comparisons (Section 3).

For the -500 to 1900 CE period, VSSI is taken from the eVolv2k reconstruction (Toohey and Sigl, 2017). These estimates are based on a set of continuous sulfate records from a suite of ice cores from Greenland and Antarctica. As in earlier ice core-based reconstructions (e.g., Gao et al., 2008), tropical eruptions are identified by simultaneous deposition in both Antarctica and Greenland, and signals present in only one hemisphere are assumed to result from extratropical eruptions. VSSI is estimated by applying empirically-derived transfer functions to the ice sheet average sulfate flux values (Toohey and Sigl, 2017, Gao et al., 2008).

Volcanic stratospheric sulfur injection estimates for the Holocene (from 9,500 BCE or 11,500 years BP to 1900 CE; comprising 1,496 eruptions) are available from the HolVol v1.1 reconstruction (Sigl et al., 2022). The method of VSSI reconstruction is very similar to that of eVolv2k, although the number of ice cores used is necessarily smaller since fewer cores cover the full





Holocene. Despite the lower number of samples, the HolVol reconstruction shows good agreement with the 2,500-year long eVolv2k record, strengthening confidence in its accuracy (Sigl et al., 2022). Uncertainty in the timing of eruptions in HolVol

is estimated to be ±1 to 5 years on average over the last 2,500 years and better than ±5 to 15 years for the rest of the Holocene. Uncertainties in the VSSI values in HolVol are typically around 35%.

Ice core derived VSSI estimates have been recently reconstructed for the 60-9 ka BP period, which covers the late glacial period as well as the Early Holocene (Lin et al., 2022). Due to the thinning of ice sheets with age, the reconstruction of Lin et al. (2022) is limited to eruptions which produced strong deposition to both Greenland and Antarctica. The reconstruction

provides estimates of stratospheric sulfate loading for 85 eruptions with bipolar deposition. To convert the sulfate aerosol mass estimates of Lin et al. (2022) to units of mass sulfur, we divide by a factor of 3 to account for the ratio of molar masses for sulfur (32 g/mol) to sulfate (96 g/mol).

### 2.3 Deriving VSSI from tephra

VSSI is estimated assuming a linear relationship between VSSI and erupted volume, i.e., a power law relationship between

VSSI and eruption magnitude, as used by Pyle et al. (1996) and Metzner et al. (2014). Here, we derive a fit of VSSI to magnitude using ice core-derived VSSI (Section 2.2) and tephra-based magnitudes (Section 2.1) for identified eruptions, as well as recent eruptions for which estimates of sulfur emission are available from satellite instruments (Carn, 2022) (Fig. 4).

appendix Table A1 lists eruption data which is used to derive a relationship between VSSI and eruption magnitude. For the period 1980 to 2014, we use sulfur emission estimates from satellite instruments compiled by Carn (2022). Emitted sulfur is

matched to the eruptions listed in LaMEVE for this period. For the period before the satellite era, we rely on VSSI estimates and eruption attributions included in the eVolv2k reconstruction (Toohey and Sigl, 2017), supplemented with a few events from the HolVol v1 reconstruction (Sigl et al., 2022). Compared to prior investigations of the VSSI-vs-magnitude relationship (Pyle et al., 1996; Metzner et al., 2014), this data compilation includes more satellite-based estimates, as well including ice core-based estimates, extending the coverage to larger magnitude eruptions, particularly due to the inclusion of recently

attributed ice core signals of Okmok (McConnel et al., 2020) and Crater Lake (Sigl et al., 2022) (Fig. 4).

### 2.4. Semi-stochastic VSSI time series generation

In order to correct for the incompleteness of the tephra-based eruption time series, we manufacture a synthetic eruption time series with the same statistics as an input data set, with timing and magnitudes of eruptions randomized. To construct our

supplementary synthetic eruption time series, we draw on the efficient methodology of Bethke et al. (2017), but using as a basis the recent HolVol ice core-based reconstruction over the years -4000 to 1900 CE. The algorithm of Bethke et al. (2017) stochastically produces a new eruption time series based on the eruption magnitude-frequency distribution of the input time series. Our time range is chosen so as to include a large period to enhance the statistical basis of the frequency distribution, while excluding the deglaciation period, for which the eruption frequency is amplified compared to background periods. When



an eruption is randomly generated, characteristics of the eruption from HolVol, including VSSI, eruption region (tropical, NH extratropical or southern hemisphere (SH) extratropical), and month are copied into the constructed synthetic time series. For eruptions that are unidentified in the base data set (which in our case is the vast majority), we randomize the eruption characteristics in the output data: the eruption month is randomized uniformly across the calendar year, and the eruption latitude is randomized within the identified tropical, NH and SH bands using the probability density of the LaMEVE data set between

10 ka BP and the present. We repeat this process 100 times to produce 100 synthetic eruption time series.

To produce an ensemble of final VSSI time series therefore requires merging each synthetic time series--based on the statistics of the HolVol ice core data--with the evidence from the tephra record. Merging the two eliminates the decreasing eruption frequency backwards in time present in the tephra record, assuming that this characteristic of the tephra record is a product of incompleteness. It also assumes that the true eruption frequency distribution is approximately static with time. For each of the

100 ensemble members, the synthetic and tephra records are merged so that each event in the tephra record is inserted into the synthetic record while also removing from the synthetic record a synthetic event with closest matching magnitude within a window of 500 years. To represent the dating uncertainty in the tephra-based events in the ensemble of forcing time series, and also avoid clumping of eruptions around intervals of 1000's of years due to the limited resolution of the reported dates of the tephra-based events, thus potentially creating an artificial millennial-scale periodicity in the radiative forcing, we add a

random, normally distributed perturbation to the reported date of the tephra-based eruptions, based on the estimated dating uncertainty (see Section 2.1 and figure 3). Thus, the dates of tephra-based eruptions in our reconstruction match the original dates for an eruption within the reported uncertainty. The dating perturbation is performed separately for each of the 100 ensemble members, so that the date assigned differs in each realization of the data product, and the spread in dates between realizations depends on the dating uncertainty of the tephra event.

**2.5. Aerosol optical properties**

The impact of volcanic aerosol radiative forcing is implemented in climate models in different levels of complexity. The simplest models take as input variations in the top of atmosphere radiative flux anomalies (W/m$^2$), which represent the net effect of scattering and absorption of radiation by stratospheric aerosol (e.g., the energy balance model used in Pages-2k-Consortium, 2019). Comprehensive climate models, on the other hand, may require detailed optical properties of the aerosol,

as a function of latitude, height, wavelength and time, to be used in the radiative calculations of the model (as e.g., for the Max Planck Institute Earth System Model used in Bader et al., 2019).

To produce timeseries of the radiative impacts of past eruptions, including the detailed optical properties required by comprehensive models, we use the Easy Volcanic Aerosol (EVA) forcing generator (Toohey et al., 2016). This simple model takes as input the eruption timing, VSSI estimates, and eruption latitude, and produces aerosol extinction, single scattering

albedo and scattering asymmetry factor as a function of latitude, altitude, wavelength and time. These variables are the result of a simple 3-box model of stratospheric transport, scaling approximations between aerosol mass and AOD at 0.55 μm, and Mie theory which describes the scattering of radiation by spheres. The overall impact of stratospheric aerosol on the Earth's



energy balance is roughly proportional to the AOD in the visible part of the electromagnetic spectrum, so it is common to illustrate the volcanic forcing simply as the AOD at 0.55 µm, either as a function of latitude, or as a global (area weighted) 270 mean.

### 2.6. A note on date formatting

Throughout the manuscript we use two conventions with regarding to dates. For periods extending no further than around 2,500 years into the past, we use a variation of the ISO8601 format, which is very similar to the usual "Common Era" (CE) system, differing though in the sense that the ISO8601 system includes a year 0, while the Common Era system does not. The 275 two systems therefore differ by one year for years before 1 CE, for example, 44 BCE would correspond to the year -43 in the ISO8601 system.

For dates further in the past, we use the widely used "Before Present" (BP) system, which indicates the number of years before 1950.

### 280 3. Results

#### 3.1 VSSI-to-eruption magnitude relationship

The relationship between eruption magnitude and VSSI as observed by satellite instruments over the most recent decades and estimated from ice cores over the last 2500 years and Holocene is shown in Figure 4. Satellite observations offer predominantly information on the VSSI from M=4 to M=5 eruptions: the only eruptions larger than M=5 that have been observed directly are 285 the 1991 Pinatubo (M=6.1) and Cerro Hudson (M=5.8). For M=4 eruptions, the VSSI observed by satellites covers a range from 0 to approximately 1 TgS. The largest values of VSSI observed from satellites are associated with the eruptions of Pinatubo (M=6.1, VSSI=7.6 TgS) and El Chichón (M=5.1, VSSI=4.0 TgS). Eruptions from the ice core reconstructions extend the range of eruption magnitudes included in the analysis, we use here 24 eruptions, of which 21 have magnitude greater or equal to 5.0.

The VSSI values from satellite instruments and ice core reconstructions taken together show a proportionality with eruption magnitude: larger eruption magnitudes lead generally to larger VSSI values. Following Pyle et al. (1996), we fit a power law relation to the data to obtain a best fit relationship for VSSI in TgS: $VSSI = (1.67 \times 10^{-5}) \times 6.27^{M}$

This fit is very similar to fits produced by Pyle et al. (1996) based solely on satellite data from the period 1979–1993, and also somewhat similar to the fit presented by Metzner et al. (2014), who based their fit on petrologically obtained sulfur emission 295 values from Central American eruptions (see Figure 4).

There is clearly a significant amount of scatter around the line of best fit. This scatter can be the result of uncertainties in the eruption magnitude estimates as well as the VSSI values, but is likely dominated by actual variability in sulfur emission between different eruptions with similar magnitude (e.g., Andres et al., 1993; Sigurdsson, 1990), and on the proportion of



emitted sulfur that is injected into the stratosphere. The strong apparent scatter for M=4 eruptions in Figure 4 represents a

modest scatter in term of absolute numbers, with VSSI ranging from approximately 0 to 1 TgS, reflecting variations in the

sulfur output as well as the modulating effect of plume heights which in many cases are not high enough to bring sulfur into

the stratosphere. For M>=5, the proportionality between VSSI and magnitude is more compact, the majority of events falling

within approximately an order of magnitude around the best fit line. Notable outliers for which the VSSI is larger than the best

fit line relationship include Laki (1783), for which the Greenland VSSI estimate has been suggested to potentially include a

significant amount of tropospheric aerosol (Lanciki et al., 2012), and Hekla (1766), another Icelandic eruption for which the

same scenario may hold. In contrast, a VSSI much smaller than expected, based on the best fit relationship, is seen for the

Millennium eruption of Changbaishan (940 CE), which has been discussed previously and may be due to some combination

of sulfur-poor magma (Horn and Schmincke, 2000) and short stratospheric lifetime due to injection height or seasonal

atmospheric dynamics (Iacovino et al., 2016). Accordingly, VSSI estimated from eruption magnitudes should be understood

to have significant uncertainty for any individual eruption.

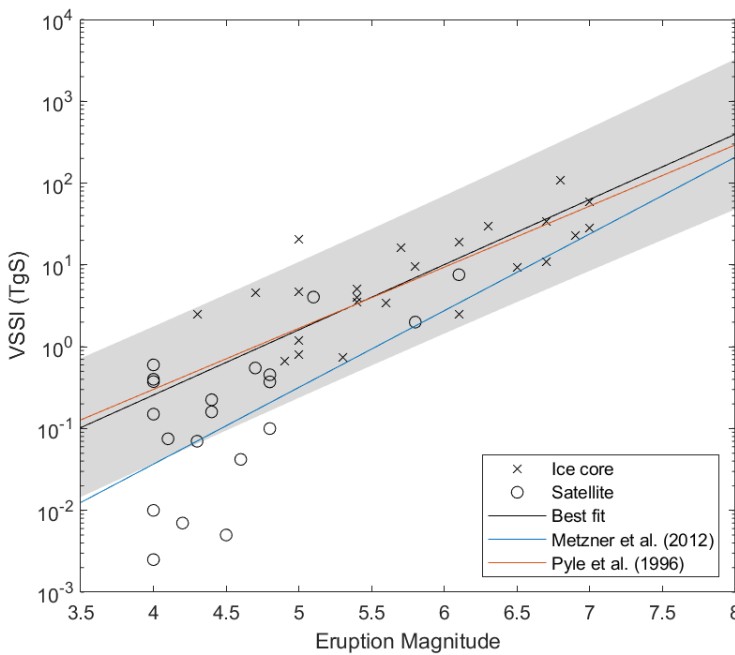

**Figure 4: Volcanic stratospheric sulfur injection (VSSI) derived from ice cores (Toohey and Sigl, 2017; Sigl et al., 2021) and satellite observations (Carn, 2022) as a function of eruption magnitude from the LaMEVE database (Crosweller et al., 2012). A least squares power law best fit is shown in black, compared to similar fits from Pyle et al. (1996) and Metzner et al. (2014). A 1-sigma uncertainty**
**range to the fit is shown as gray shading.**



### 3.2 Common Era

During the Common Era (0-2000 CE) the eruption rate as recorded in the tephra records is larger than that of the ice core
records during the chosen base period (-4000 to 1400 CE). Therefore, we can base the PalVol reconstruction purely on the
tephra records without any synthetic events that could otherwise be included to compensate for undersampling by the tephra
(Fig. 5).

The timeseries of VSSI calculated from the tephra magnitude estimates shows reasonable agreement with the ice core-based
eVolv2k reconstruction (Fig. 5). The Rinjani (Samalas) eruption of 1257 CE, which produced the largest VSSI in eVolv2k
(59.4 TgS) is well reproduced in the tephra time series, with an estimated VSSI of 69.4 TgS. The large eruptions of Tambora
(1815) and Krakatau (1883) are apparent in both time series, albeit larger by a factor of approximately two in the tephra-based
estimates compared to eVolv2k. The tephra record includes a large VSSI associated with the Changbaishan eruption of 942,
which is not found in ice cores, which is likely due to the sulfur-poor content of the erupted magma (Horn and Schmincke,
2000). Other prominent events in the 1st millennium CE of the tephra VSSI time series, including TBJ (450±30 CE, VSSI=38.9
TgS), Taupo (230±16 CE, VSSI=57.2 TgS) and Ambrym (190±135, VSSI=47.2 TgS) are likely to be responsible for ice core
signals within the dating uncertainty of the tephra events.

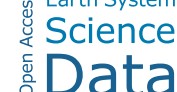

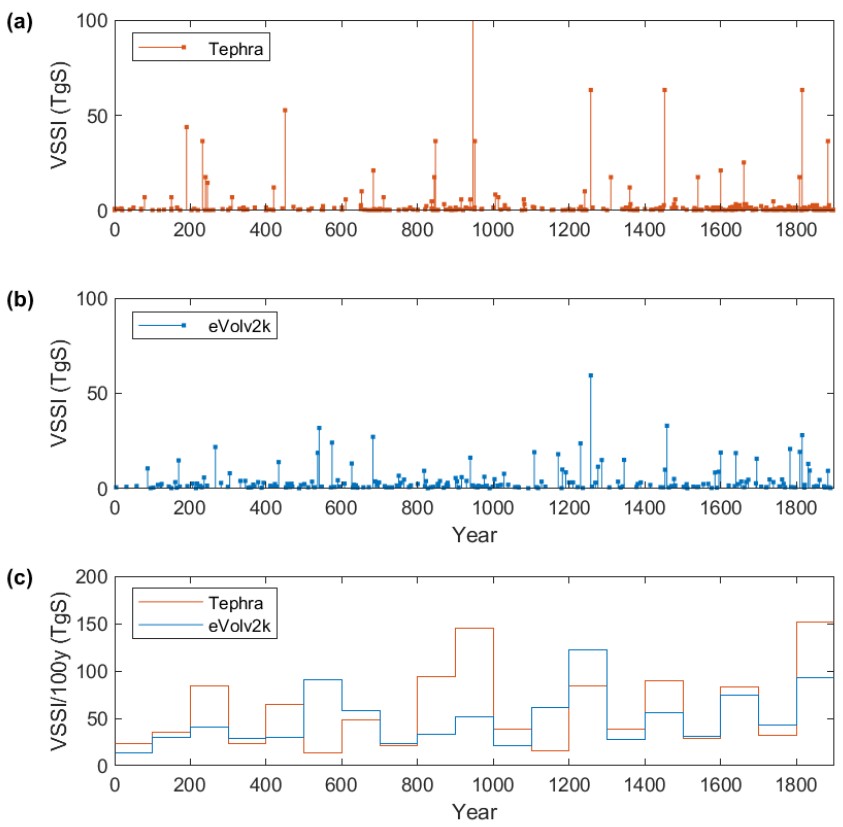

**Figure 5: VSSI estimates for the Common Era from (a) tephra and (b) ice cores (eVolv2k, Toohey and Sigl, 2017). (c) Centennial total VSSI for both data sets, with the correlation of the two centennial time series.**

Given the large uncertainties in VSSI estimated from tephra (and the non-negligible uncertainties in the same value from ice cores), we would not and should not expect the VSSI values for individual eruptions to agree to a precision better than an order of magnitude. When multiple eruptions are averaged together, we expect the errors to compensate to some degree, and cumulative VSSI values to become more reliable. Centennial total VSSI is shown for the tephra reconstruction and eVolv2k in Figure 5c. The correlation between the two centennial time series is notable, with an overall correlation coefficient of

R=0.50. The correlation is especially good in the 2nd millennium, with both reconstructions showing elevated mean VSSI amounts for the 13th and 19th centuries, as well as more modest elevated values for the 15th and 17th centuries. While the tephra time series misses the elevated VSSI values of the 6th Century, there is some agreement in the elevated VSSIs during the 2nd and 3rd centuries.

### 3.3 Holocene

VSSI reconstructed from tephra is shown in Figure 6 for the Holocene period (roughly 11 to 0 ka BP) and compared to the

HolVol ice core-based reconstruction. Both data sets show an increase is large magnitude VSSI in the early Holocene, around

11 to 7 ka BP. Particularly, both data sets include four quite large events between 9 and 7 ka BP, with the tephra-based values

reaching approximately 100 TgS while the ice core values reach up to over 150 TgS.

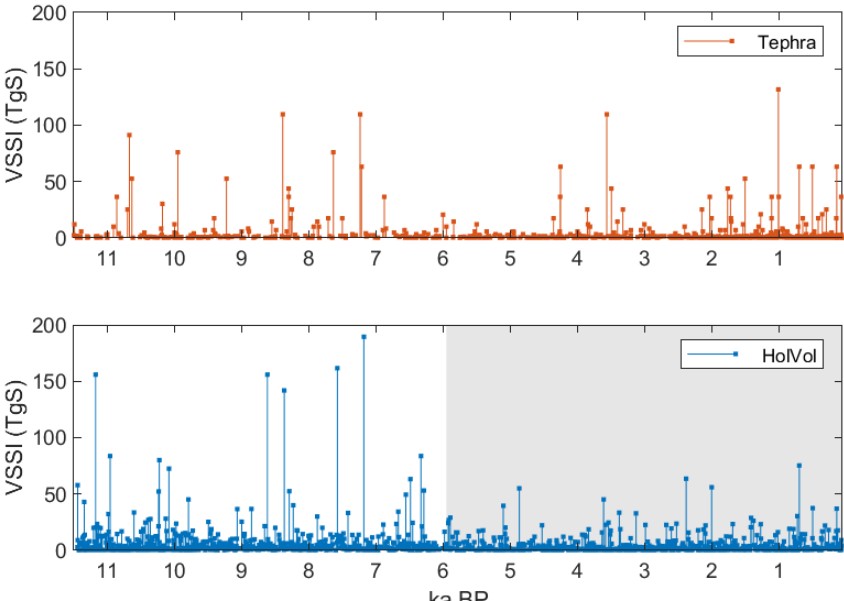


**Figure 6: Comparing tephra and ice core-based VSSI time series over the Holocene. Upper panel: VSSI time series derived from tephra records. Lower panel: HolVol ice core-based VSSI time series (Sigl et al., 2022). Gray shading shows the period used to base eruption statistics for the synthetic time series.**

In a next step we compare the statistical characteristics of the tephra data, the PalVol reconstruction ensemble, and the HolVol

ice core-based VSSI reconstruction over the period of the Holocene on millennial timescales. Comparing first the number of

events in the tephra and ice core data sets (Fig. 7a) we see that the number of tephra events drops rapidly with increasing age,

from ~275 events/ka from 0-1 ka BP to less than 100 events/ka before 2 ka BP, to less than 50/ka in the early Holocene,

consistent with prior analyses of the LaMEVE database (Brown et al. 2014).

From roughly 0 to 9 ka BP, the ice core number of events holds approximately steady at around 100 events per ka, which

increases during the deglaciation period, reaching a maximum of ~160 events per ka around 11 ka BP. In the most recent 2 ka,

the frequency of tephra events is larger than the frequency of ice core events. This is likely due in part to overcounting of

tephra events, due to overestimation of the magnitude of events (increasing the absolute number of VEI>=4 events included

in LaMEVE). We speculate that another source of discrepancy may be undercounting of VEI>=4 eruptions by ice cores, since

not all eruptions may leave traces in ice cores if not explosive enough or sulfur poor. Before 0 CE, the incompleteness of the



tephra record is clear compared to the ice cores, and the degree of incompleteness increases roughly linearly until around 6 ka BP. There may be a weak local maximum in tephra events around 9 to 10 ka BP CE in agreement with the ice core increase here, but the difference in tephra events between this period and the local minimum at 7 to 8 ka BP is rather small.

Next, we compare the cumulative VSSI per millennium derived directly from the tephra data compared to the HolVol VSSI database (Fig. 7b). The tephra-derived VSSI per ka is, in all millennia before 2 ka BP, smaller than the HolVol values, which

is perhaps unsurprising given the much smaller number of events in the tephra database compared to the ice cores. In contrast, over the Common Era (0-2 ka BP), the tephra VSSI/ka is larger or roughly equal to the ice core VSSI/ka.

Over the Holocene, the average VSSI per millennium from HolVol is 638 TgS, while that from the pure tephra data is 231TgS per millennium.

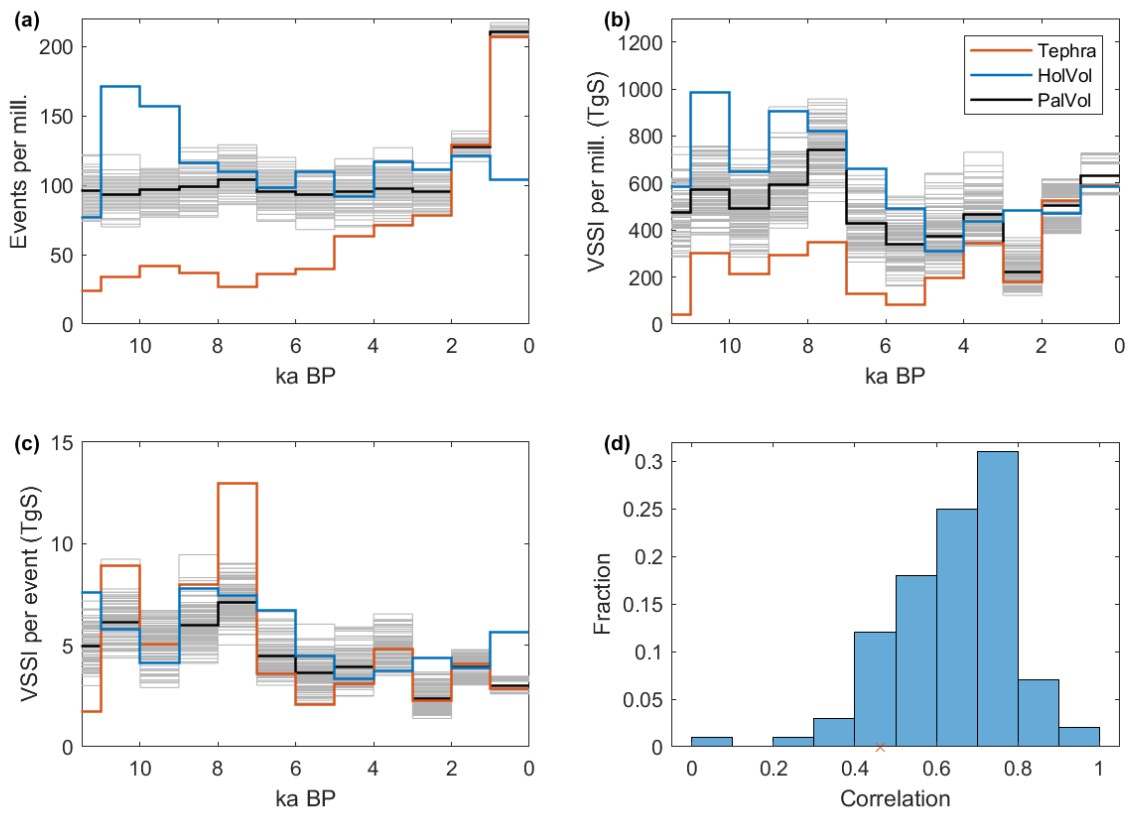


**Figure 7: Millennial-scale variations in eruptive characteristics over the Holocene in ice core (HolVol), tephra and PalVol time series. (a) Number of volcanic events per millennium in each dataset, (b) cumulative VSSI per millennium, (c) average VSSI per event, (d) probability density function of correlation coefficient calculated between the ensemble of semi-synthetic millennial cumulative VSSI time series with the HolVol time series. In (d), the correlation between the tephra time series and HolVol is indicated by the orange**
**cross. In panels a-c, gray lines indicate values for each of the 100 stochastic realizations, and the black line the ensemble mean.**





Despite clear differences between the tephra and ice core VSSI reconstructions for specific events, and a bias with tephra showing lower values for most millennia of the Holocene, there is correlation between the two datasets in terms of millennial VSSI over the Holocene, with both showing larger VSSI/ka values in the early Holocene (11 to 7 ka BP) compared to the mid

Holocene (2 to 7 ka BP). The correlation coefficient between the millennial VSSI totals for the tephra and ice core-based datasets over the 10 to 2 ka BP CE period is R=0.46. This correlation comes about despite the strong undercounting of the tephra database in the early Holocene. Evidently, although the number of events captured by the tephra data sets is small, the events that are counted tend to be the larger eruptions, which contribute the most to the VSSI millennial sums. Figure 7c shows the average VSSI per event as a function of millennium, which shows a similar structure for both the tephra and ice core data,

with larger average VSSI per event in the early Holocene compared to the mid and late Holocene. This implies that the increase in VSSI/ka in the early Holocene arises from an increase in the frequency of large eruptions, which is evident in both the ice core and tephra data.

Now we compare the semi-stochastic PalVol VSSI reconstruction, constructed by adding stochastically generated events to the tephra data, to the statistics of the HolVol reconstruction. By construction, the PalVol reconstruction includes a relatively

constant number of events per millennium, around 100 events/ka (Fig. 7a), in agreement with the frequency of events in the HolVol reconstruction over the chosen base period -4000 to 1900 CE (roughly 6 to 0 ka BP). The number of events/ka for each individual ensemble member of the reconstruction will vary around this number based on the stochastic event generation, and the inter-millennial variance matches well with the variance of the HolVol reconstruction over the base period. The PalVol reconstruction clearly does not include the increase in events/ka in the early Holocene that is seen in HolVol (Fig. 7a). This is

by construction, since we do not adjust the eruption frequency probability with time.

Nonetheless, in Figure 7b, we see that after adding the stochastically generated events to create the PalVol forcing reconstruction, the agreement of the millennial distribution of VSSI/ka with the HolVol reconstruction improves compared to the pure tephra time series. First, the overall bias is reduced (but not eliminated), with an average VSSI/ka of 505 TgS for PalVol compared to 639 TgS for HolVol. Secondly, since more stochastically generated events are added to the early Holocene

compared to the late Holocene to make up for the larger bias in event frequency (see Fig. 7a), the VSSI/ka is boosted more in the early Holocene. This improves the correlation: the mean PalVol VSSI/ka time series shows a correlation of R=0.73 with HolVol. A histogram of the correlation coefficient between each individual PalVol VSSI time series with HolVol (Fig. 7d) shows the largest proportion of ensemble members have a correlation of 0.7-0.8. We conclude that over the Holocene, the addition of stochastic events to the tephra data improves resulting time series in comparison to ice core derived reconstructions,

both by reducing the low bias in the tephra data, and improving the inter-millennial variability of VSSI.

### 3.3 Last glacial cycle

The PalVol VSSI reconstruction is compared to the ice core-based VSSI reconstruction of Lin et al. (2021) in Figure 8. The Lin et al. (2021) events used here constitute 85 eruptions with matched bipolar signals with deposition of >20 kg/km² in Antarctica and >10 kg/km² in Greenland between 60–9 ka BP. Tephra events in this period have a median dating uncertainty





of 2300 years (900-4,600 years 0.25-0.75 interquartile range), therefore, we don't expect clear temporal matches to the ice
core events for the reported dates of tephra events. Despite this, we find a decent agreement in the VSSI frequency distribution
of the largest events in both time series for the period overall. For example, we find 11 events with VSSI >100 TgS in the Lin
et al. (2021) time series, while in the PalVol record for the same 60-9 ka BP period, we have 10 such events. The largest VSSI
signal in the PalVol reconstruction over this period is associated with the 27 ka BP Taupo Oruanui eruption with M=8.1,

leading to an estimated VSSI of 480 TgS, a factor of almost 5 times greater than the ice core derived value of 127 TgS. This
represents an overestimate compared to the ice core-based reconstruction for this event, which is not overly surprising given
the uncertainty in the tephra-based reconstruction method.

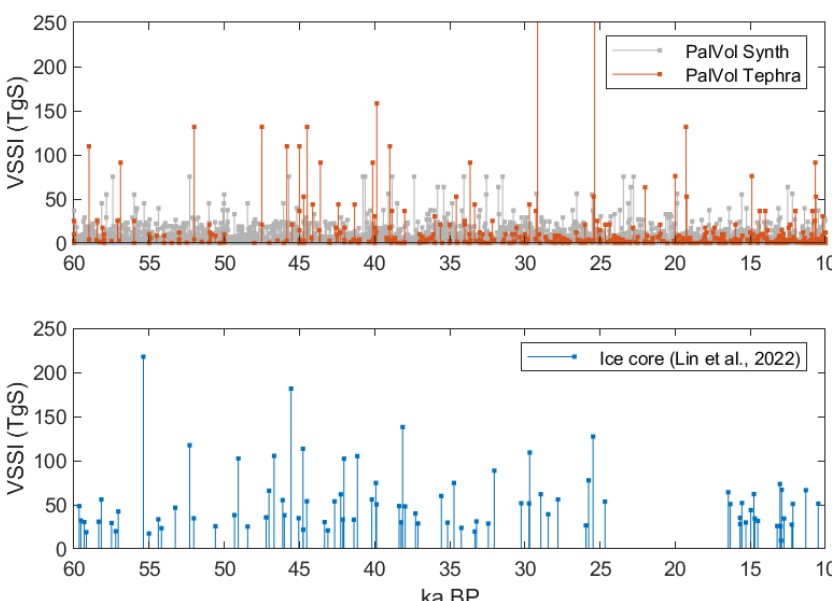

**Figure 8: Volcanic stratospheric sulfur injection from PalVol (top) and the ice core-based reconstruction of Lin et al., 2022, for the 60-9 ka BP period. Stochastically generated PalVol VSSI (in gray) are from a single ensemble member, while tephra-based estimates (red) are shown at the reported dates (not randomized).**

Reconstructed VSSI is shown in Figure 9 for the full PalVol reconstruction period. Individual eruptions are shown in panels a
and b, for which the decreasing number of detected eruptions with age, and the resulting increasing number of synthetic

eruptions included is apparent. The four largest VSSI estimates in PalVol exceed the range plotted on Figure 9: the
corresponding VSSI values are listed in Table 1 as part of the top twenty VSSI estimates. The eruption of Toba is the largest
eruption of the past 140,000 years, and results in an estimated 3,000 TgS VSSI using our method, with a wide estimated
uncertainty range spanning 310 to 29,000 TgS. Prior estimates of Toba's sulfate emission fall within a very broad range,
multiple studies were compiled by Oppenheimer (2002) to define a range of 35 to 3,300 TgS. More recently, Costa et al. (2014)

estimated Toba's sulfur emission as 850-1,750 TgS, which falls within our uncertainty range, while Crick et al., (2021)

estimated a range of 72-233 TgS, which falls outside our uncertainty range. It must be stressed that our estimates of VSSI for

the strongest eruptions is based on an extrapolation of the VSSI-to-magnitude relationship beyond what has been observed,

and non-linearities in the physical processes (e.g., plume collapse) are not considered here.


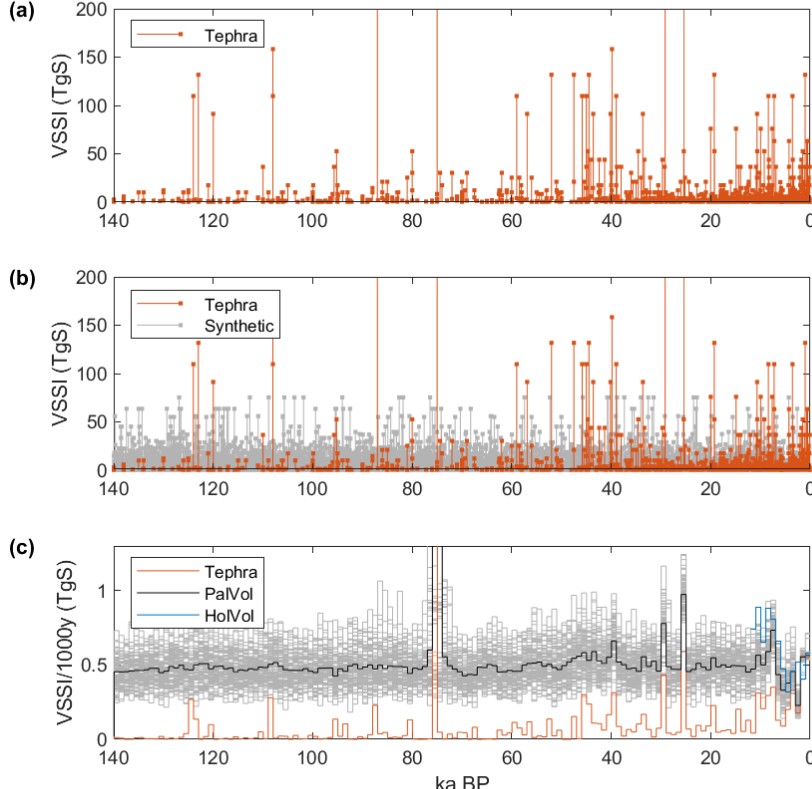

**Figure 9: Tephra-based volcanic stratospheric sulfur injection (VSSI) reconstruction. (a) VSSI based on the pure tephra record. (b) A semi-synthetic VSSI time series, based on merging a stochastic synthetic record with the tephra reconstruction. (c) Millennial VSSI rates for the tephra, TephraSynth and HolVol reconstructions.**

The Los Chocoyos eruption of Atitlan, ~75 ka BP, has the same magnitude as the Taupo eruption, and therefore the same

estimated VSSI of 480 TgS with a range of 57 to 4,000 TgS. This value is very similar to the 343 TgS estimated by Metzner

et al. (2014), although the method used there was very similar to that used here.

Our estimate for the Changbaishan eruption (946 CE) is 130 TgS, while Horn and Schmincke (2000) estimated a release of

5.7 TgS, and Iacovino et al. (2016) a release of 45 TgS. A lack of strong ice core sulfate signals around the documented time

of the eruption has been used as evidence that the sulfur emission from Changbaishan must have been quite minor

(Oppenheimer et al., 2017), and likely much lower than the estimate here based only on the eruption magnitude.



**Table 1: Estimated VSSI for the top 20 largest eruptions of the past 140,000 years based on the LaMEVE database.**

| Volcano | Year | Year BP | Magnitude | VSSI (TgS) | Min VSSI (TgS) | Max VSSI (TgS) |
|---|---|---|---|---|---|---|
| Toba | -73050 | 75000 | 9.1 | 3000 | 310 | 29000 |
| Taupo | -23410 | 25360 | 8.1 | 480 | 57 | 4000 |
| Atitlán | -73050 | 75000 | 8.1 | 480 | 57 | 4000 |
| Aira | -27205 | 29155 | 7.9 | 330 | 40 | 2700 |
| Asosan | -85050 | 87000 | 7.7 | 230 | 29 | 1800 |
| Campi Flegrei | -37900 | 39850 | 7.5 | 160 | 20 | 1200 |
| Ata | -106050 | 108000 | 7.5 | 160 | 20 | 1200 |
| Pacaya | -122050 | 124000 | 7.5 | 160 | 20 | 1200 |
| Changbaishan | 946 | 1004 | 7.4 | 130 | 17 | 1000 |
| Long Island | -17328 | 19278 | 7.4 | 130 | 17 | 1000 |
| Opala | -42534 | 44484 | 7.4 | 130 | 17 | 1000 |
| Okataina | -45550 | 47500 | 7.4 | 130 | 17 | 1000 |
| Maninjau | -50050 | 52000 | 7.4 | 130 | 17 | 1000 |
| Asosan | -121050 | 123000 | 7.4 | 130 | 17 | 1000 |
| Santorini | -1610 | 3560 | 7.3 | 110 | 14 | 840 |
| Kikai | -5284 | 7234 | 7.3 | 110 | 14 | 840 |
| Kurile Lake | -6437 | 8387 | 7.3 | 110 | 14 | 840 |
| Gorely | -37031 | 38981 | 7.3 | 110 | 14 | 840 |





| Nemo Peak | -43050 | 45000 | 7.3 | 110 | 14 | 840 |
|-----------|--------|-------|-----|-----|----|-----|
| Shikotsu | -43883 | 45833 | 7.3 | 110 | 14 | 840 |

Figure 9c shows the millennial average VSSI for the PalVol reconstruction as well as the pure tephra data and the HolVol ice core reconstruction. Each PalVol ensemble member is shown in gray: these reconstructions generally show millennial average VSSI ranging from approximately 0.3 to 0.7 TgS/year, consistent with the mean value and variability from HolVol. Exceptions occur in millennia which contain the strongest eruptions, for which the millennial average VSSI can increase by a factor of 2 or more. The ensemble average PalVol millennial VSSI is approximately constant with age except for the perturbations due to

the largest eruptions, and some small increases due to enhanced numbers of detected strong eruptions, for example in the 55-40 ka BP period.

    One realization of PalVol VSSI time series is visualized in Figure 10, with VSSI magnitudes plotted as "bubbles" as a function of ka BP and latitude. Many characteristics of the LaMEVE database are apparent: the decreasing eruption sampling with increasing age, the dependence of this sampling with latitude, with a generally more complete sampling in the NH mid

latitudes. This figure illustrates the distribution of the tephra-based and synthetic eruptions in time and latitude, and specifically the latitude distribution of the synthetic events based on the latitudinal distribution of the LaMEVE eruptions.

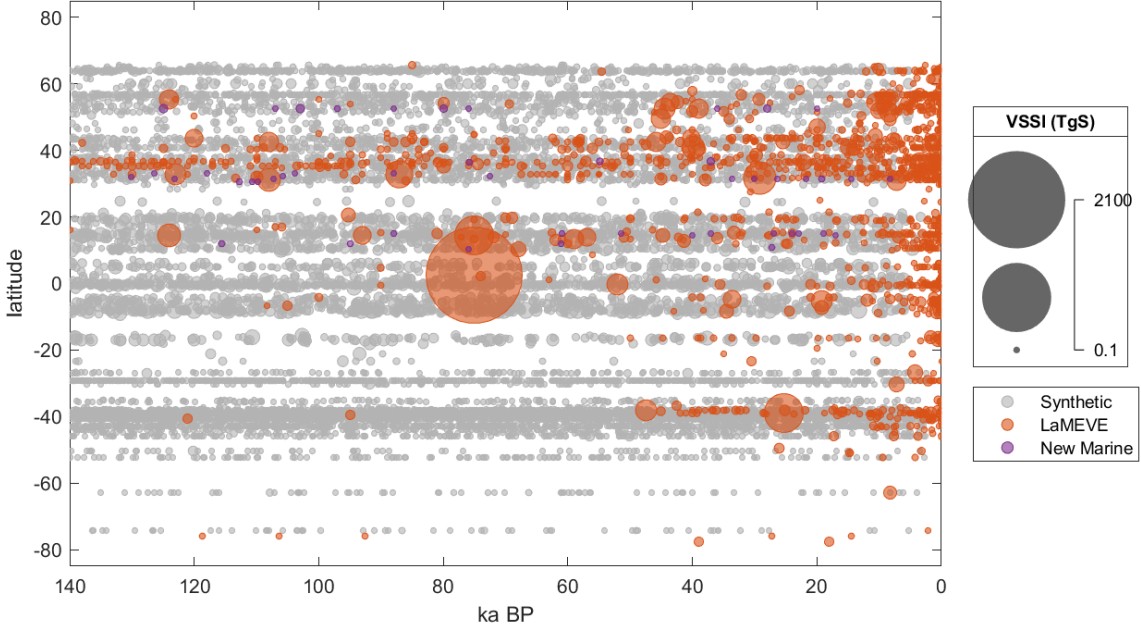

**Figure 10: The PalVol global VSSI product, with the coordinates of each circle representing the time and latitude of an eruption, and the circle size the VSSI in TgS. Circle colors signify the type of event, either detected events compiled in LaMEVE (red) or**

**additional records from marine sediment cores (purple), or the stochastic synthetic events used here to fill the record (gray).**



**Data availability**

The PalVol volcanic stratospheric sulfur injection stratospheric aerosol optical depth datasets described herein are available through the World Data Center for Climate in netCDF format (https://doi.org/10.26050/WDCC/PalVolv1; Toohey and Schindlbeck-Belo, 2023).

**4. Conclusions and Discussion**

We have produced PalVol VSSI reconstructions that cover the last glacial cycle (the last 130 kyrs). The ensembles incorporate tephra data of past eruptions, as well as stochastically generated events to attempt to correct for the undersampling of events with increasing age. VSSI values are derived from the eruption magnitude, which is itself estimated based on the thickness of tephra layers in the surrounding of a volcano. Importantly, the VSSI values for individual eruptions have significant

uncertainties, as we (and prior works; e.g., Andres et al., 1993; Sigurdsson, 1990) showed that the amount of sulfur released by an eruption can vary by orders of magnitude for any given eruption magnitude (Fig. 4). Especially for M=4 eruptions the VSSI varies significantly since some reach the stratosphere and some do not (Fig. 4), which depends on their actual plume height, but also on latitude. Uncertainties in VSSI are included in our reconstruction as upper and lower bounds on the VSSI for each eruption, based on the uncertainty in our derived magnitude to VSSI relationship. We note as well that this relationship

may not be constant in time, and indeed changes may occur, e.g., to changes in the height of the tropopause or in stratospheric circulation from glacial conditions to interglacials. This would be similar to predicted changes in future volcanic radiative forcing due to changes in the tropopause height due to climate change (Aubry et al., 2016). This is something we did not take into account in our reconstructions. Nonetheless, although the VSSI for any specific eruption is quite uncertain, we expect that due to the observed relationship between VSSI and magnitude, this will be compensated when averaged over a sufficiently

large set of eruptions, and therefore the tephra data may contain information on variations in VSSI on long time scales.

Furthermore, the PalVol reconstruction shows good agreement with the ice core based HolVol VSSI time series in terms of millennial variations in cumulative VSSI, but not particularly in terms of the timing and magnitude of individual events.

This agreement arises from the shared increase in cumulative VSSI in the early Holocene compared to the mid and late Holocene. In both data sets, this arises from the increased frequency of relatively large eruptions. Evidently, while the tephra

time series is incomplete during this period, the sampling of large events is good enough to detect the increase in the amount of large eruptions, which translate into increases in millennial cumulative VSSI.

We assumed a constant eruption frequency distribution, but there is evidence that variations in eruption frequency are driven by changes in mass distribution of ice sheets and respective sea level changes. The PalVol reconstruction includes a small increase in VSSI/ka in the early Holocene, qualitatively consistent with observations from the ice cores, but with an amplitude

which is actually smaller. Therefore, we propose that future iteration of stochastic forcing could take into account such variations, for example by using sea level reconstructions as a basis to estimate variations in eruption frequency.





The PalVol reconstruction takes the form of an ensemble of realizations, where each realization differs in terms of timing and size of the stochastically generated events, and the timing of the events taken from the tephra corresponding to the uncertainty in the tephra dating. While any single realization is unlikely to be a very accurate reconstruction of the true history of VSSI, it

is quite possible that some realizations include aspects that are realistic. Therefore, the ensemble of VSSI time series represents a probability distribution of the probable forcing from volcanic eruptions over this period, given the information we currently have from proxies.

We emphasize that users wanting the most accurate reconstruction of VSSI over the last glacial cycle could consider using a merged product, for example by concatenating the HolVol ice core time series with PalVol for the period, which occurs before

the beginning of HolVol. Future improvements to the PalVol reconstruction will be possible with the addition of new tephra data, improvements in the dating, and magnitude estimates of volcanic events.

**Appendices**

Table A1: Compilation of matched eruption magnitudes from LaMEVE with volcanic stratospheric sulfur injection (VSSI) estimates from satellite measurements or ice core-based reconstructions. Volcano. eruption year and magnitude taken from the

LaMEVE database. VSSI in TgS is taken from 3 sources. indicated by last column: 1: from satellite observations compiled by Carn et al. (2022); 2: the eVolv2k ice core reconstruction (Toohey and Sigl, 2017); 3: the HolVol ice core reconstruction (Sigl et al,. 2022).

| Volcano | Eruption Year | Magnitude | VSSI (TgS) | Source |
|---|---|---|---|---|
| Puyehue-Cordón Caulle | 2011 | 4.8 | 0.1 | 1 |
| Eyjafjallajökull | 2010 | 4 | 0.0025 | 1 |
| Merapi | 2010 | 4 | 0.15 | 1 |
| Sarychev Peak | 2009 | 4 | 0.6 | 1 |
| Chaitén | 2008 | 4.2 | 0.007 | 1 |
| Reventador | 2002 | 4.6 | 0.042 | 1 |
| Láscar | 1993 | 4.4 | 0.225 | 1 |
| Spurr | 1992 | 4 | 0.4 | 1 |
| Hudson. Cerro | 1991 | 5.8 | 2 | 1 |
| Pinatubo | 1991 | 6.1 | 7.597 | 1 |
| Kelut | 1990 | 4.1 | 0.075 | 1 |




| Redoubt | 1989 | 4.3 | 0.07 | 1 |
|---|---|---|---|---|
| Augustine | 1986 | 4 | 0.01 | 1 |
| Chikurachki | 1986 | 4 | 0.375 | 1 |
| Kliuchevskoi | 1986 | 4.5 | 0.005 | 1 |
| Chichón. El | 1982 | 5.1 | 4.045 | 1 |
| Galunggung | 1982 | 4.8 | 0.3725 | 1 |
| Alaid | 1981 | 4.7 | 0.55 | 1 |
| Pagan | 1981 | 4.4 | 0.16 | 1 |
| St. Helens | 1980 | 4.8 | 0.455 | 1 |
| Okataina | 1886 | 5.3 | 0.74 | 2 |
| Krakatau | 1883 | 6.5 | 9.34 | 2 |
| Askja | 1875 | 4.9 | 0.67 | 2 |
| Makian | 1861 | 4.7 | 4.53 | 2 |
| Cosigüina | 1835 | 5.8 | 9.48 | 2 |
| Tambora | 1812 | 7 | 28.08 | 2 |
| Grímsvötn | 1783 | 5 | 20.81 | 2 |
| Hekla | 1766 | 4.3 | 2.52 | 2 |
| Katla | 1755 | 5 | 1.18 | 2 |
| Shikotsu | 1739 | 5.6 | 3.44 | 2 |
| Katla | 1721 | 5 | 0.81 | 2 |
| Gamkonora | 1673 | 5 | 4.67 | 2 |
| Shikotsu | 1667 | 5.4 | 3.48 | 2 |
| Huaynaputina | 1600 | 6.1 | 18.85 | 2 |
| Bárdarbunga | 1477 | 5.4 | 5.12 | 2 |
| Rinjani | 1257 | 7 | 59.42 | 2 |
| Katla | 934 | 5.7 | 16.23 | 2 |
| Bárdarbunga | 870 | 5.4 | 3.99 | 2 |
| Churchill | 847 | 6.1 | 2.48 | 2 |
| Ilopango | 450 | 6.7 | 10.91 | 3 |
| Okmok | -76 | 6.7 | 33.91 | 3 |





| Aniakchak | -1645 | 6.9 | 22.77 | 3 |
| Crater Lake | -5677 | 6.8 | 107.9 | 3 |
| Khangar | -5699 | 6.3 | 30.05 | 3 |

Table A2: Additional data from a suite of marine cores and recent studies including cores samples from Central America, offshore the Izu-Bonin volcanic arc, offshore the Cape Verdes, offshore the Kamchatka peninsula.

| Sample name | Age [Ma] | Source | Tephra Volume [km³] | DRE [km³] | Magnitude |
|---|---|---|---|---|---|
| **Schindlbeck et al., 2016** | | | | | |
| 344-U1413B-1H-1-9-11 | 0.0011 | Rincon de la Vieja tephra | 0.85 | 0.40 | 4.92 |
| 334-U1378B-3H-6_29-30 | 0.027 | Rincon | 2.56 | 1.19 | 5.40 |
| 344-U1413A-1H-4-16-18 | 0.033 | Poas | 2.04 | 0.95 | 5.20 |
| 334-U1378B-4H-5_51-53 | 0.036 | Ilopango, TB4 | | | |
| 344-U1381C-2H-1-97-99 | 0.06 | Fontana | | | |
| 344-U1381C-2H-1-105-107 | 0.06 | Las Sierras | 2.02 | 0.94 | 5.30 |
| 170-1039B-2H-3-62-68 | 0.087 | Poas Lapilli Tuff | 1.67 | 0.78 | 5.20 |
| 334-U1378B-5H-4_106-108 | 0.076 | Poas Plantanar | 0.32 | 0.15 | 4.50 |
| 344-U1381C-2H-3-39-41 | 0.09 | Las Sierras | 5.05 | 2.36 | 5.40 |
| 334-U1378B-6H-4-93-95 | 0.116 | Apoyo | 7.66 | 3.58 | 5.50 |
| | | | | | |
| **Derkachev et al., 2020** | **Age [ka]** | **Source** | **Tephra Volume [km³]** | **DRE [km³]** | **Magnitude** |
| WP1† | 8.7 | Karymskyc caldera | 0.63 | 0.29 | 4.80 |
| WPL1 | ~20 | ? | 0.16 | 0.07 | 4.10 |
| WP2 | ~28 | EVF | 9.44 | 4.41 | 5.90 |
| WP3 | ~36 | EVF | 0.63 | 0.29 | 4.80 |
| WP4 | ~39 | Gorely | 7.69 | 3.59 | 5.70 |
| WPL2 | ~76 | Opala? | 0.70 | 0.33 | 4.80 |
| WP5 | ~80 | Gorely | 10.49 | 4.90 | 6.00 |
| WPL3 | ~88 | ? | 0.39 | 0.18 | 4.60 |



| | Age [Ma] | Source | | | Magnitude |
|---|---|---|---|---|---|
| WP6 | ~97 | Opala? | 2.24 | 1.04 | 5.40 |
| WP7 | ~103 | EVF | 14.31 | 6.68 | 6.20 |
| WP8 | ~107 | EVF | 1.26 | 0.59 | 5.10 |
| WP9 | ~125 | Gorely? | 14.68 | 6.86 | 6.20 |
| *volumes are calculated for a 30° single isopach following the methods described in Schindlbeck et al., 2018a | | | | | |
| | **Age [Ma]** | **Source** | | | **Magnitude** |
| **Schindlbeck et al., 2018a** | | | | | |
| 350-U1436A-1H-1-48-50 | 0.0083 | Sumisu Knoll? | | | 4.00 |
| 350-U1436A-1H-1-84-86 | 0.0145 | Sumisu/Agoashima? | | | 4.30 |
| 350-U1436A-1H-1-124-126 | 0.0217 | Sumisu/Agoashima? | | | 4.60 |
| 350-U1436A-1H-2-24-27 | 0.03 | Sumisu Knoll? | | | 4.00 |
| 350-U1436A-1H-2-42-44 | 0.0337 | Sumisu Knoll? | | | 4.50 |
| 350-U1436A-1H-3-79-81 | 0.0725 | Agoashima? | | | 5.00 |
| 350-U1436A-2H-1-25-27 | 0.088 | Hachijojima? | | | 4.40 |
| 350-U1436A-2H-2-16-18 | 0.1039 | Hachijojima/Sumisu/Agoas | | | 4.60 |
| 350-U1436A-2H-2-46-48 | 0.1058 | Agoashima/Hajijojima? | | | 4.90 |
| 350-U1436A-2H-2-68-70 | 0.1072 | Sumisu Knoll? | | | 4.20 |
| 350-U1436A-2H-3-90-92 | 0.118 | Hachijojima/Torishima? | | | 4.90 |
| 350-U1437B-1H-2-37-38 | 0.0193 | Sumisu Knoll? | | | 5.50 |
| 350-U1437B-1H-3-18-20 | 0.0263 | Sumisu Knoll? | | | 5.00 |
| 350-U1437B-2H-4-46-48 | 0.1098 | Torishima/Hachijojima? Zoned | | | 5.20 |
| 350-U1437B-2H-4-72-74 | 0.1107 | Torishima/Hachijojima? | | | 5.20 |
| 350-U1437B-2H-4-128-130 | 0.1127 | Torishima/Hachijojima? | | | 5.50 |
| 350-U1437B-2H-5-117-119 | 0.1232 | Sumisu Knoll? | | | 4.70 |
| 350-U1437B-2H-5-144-145 | 0.1264 | Minami Hachijo? | | | 4.70 |
| 350-U1437B-2H-6-25-27 | 0.1301 | Myojin Knoll | | | 4.60 |



| Eisele et al., 2015 | Age [ka] | Source | | DRE [km³] | Magnitude |
|---|---|---|---|---|---|
| C1 | 17 | Cadamosto | | 0.21 | 4.30 |
| C2 | 18-20 | Fogo | | | 5.00 |
| C3 | 21-28 | Fogo | | | 5.00 |
| C4 | 24 | Fogo | | 1.44 | 5.40 |
| C5 | 26-28 | Fogo | | | 5.00 |
| C6 | 37 | Fogo | | | 5.00 |
| C7 | 40 | Cadamosto | | 0.12 | 4.20 |
| C8 | 50-53 | Fogo | | | 5.00 |
| C9 | 61 | Fogo | | 0.97 | 5.30 |
| C10 | 88 | Fogo | | 0.95 | 5.30 |

**Author contributions**

JCSB, MT, MJ, SK, KR: conceptualization; JCSB, MT: writing the original draft; MJ, SK, KR: writing- reviewing and editing; JCSB, MT: data and storage; MT: data analyses; MT, KR, SK: funding acquisition

**Competing interests**

The authors declare that they have no conflict of interest.

**Acknowledgements**

Research has been funded by the PalMod project (www.palmod.de, subproject no. 01LP1926E). This work benefitted greatly as a result of the authors' participation in the Past Global Changes (PAGES) Volcanic Impacts on Climate and Society (VICS)
working group which in turn received support from the Swiss Academy of Sciences and the Chinese Academy of Sciences. We would like to thank the DKRZ for their help with publishing the data.

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
