# Peer review of "PalVol v1: A proxy-based semi-stochastic ensemble reconstruction of volcanic stratospheric sulfur injection for the last glacial cycle (140,000 – 50 BP)"

_Earth System Science Data, 2023_

## Community Comment (CC1)

I would like to add a short comment to this paper. In the absence of a continuous dataset of stratospheric volcanic sulfur emission and aerosol optical properties datasets prior to 11,500 years BP, I think it is a good idea to get comparable information from a synthetic product that takes into account ice core data but also tephra data from proximal deposits. I also acknowledge, that you are emphasizing inherent limitations of this synthetic product (e.g. comparably large error bars for age and strength of VSSI), and suggest to use or combine with, where possible, more precise data derived from ice-core records (Lin et al., 2022; Sigl et al., 2022; Toohey and Sigl, 2017). However, user of the dataset will likely employ it for many purposes, including linking the eruption record to rapid climate change, extreme events and societal collapse. A lot of progress has been achieved in the past decade regarding the dating of past eruptions using for example new-generation ice-core analyses or annual-resolution radiocarbon analyses. The LaMEVE database which forms the backbone of the PalVol v1 reconstruction, however, was built over a decade ago (Crosweller et al., 2012), and represents an earlier state of research, which has since been modified or refined many times.

Upon noting numerous eruption ages in Table A1 which differ from what is currently considered the best estimates, I have downloaded the LaMEVE eruption catalogue (https://www2.bgs.ac.uk/vogripa/view/controller.cfc?method=lameve) and critically assessed the dating of past eruptions in this dataset against revised age estimates published since. My analysis is incomplete and biased towards Late Glacial, Holocene and Common Era eruptions. Dating errors are considered to represent 2σ uncertainties unless stated otherwise.

| Volcano | Date (PALVOL) | Date (revised) | Method for revision | References |
|---------|---------------|----------------|---------------------|------------|
| Changbaishan | 942 (±4) CE | Nov 946 CE | Dendrochronology, ice core, radiocarbon | (Oppenheimer et al., 2017) |
| Katla (Eldgjá) | 934 (±2) CE | 939 CE | Documentary, ice core | (Oppenheimer et al., 2018; Sigl et al., 2015) |
| Bárdarbunga (Vatnaoldur) | 870 CE | 877 (±1) CE | Ice core | (Plunkett et al., 2023) |
| Churchill (White River Ash, eastern lobe (WRAe) | 847 (±1) CE | 853 (±1) CE | Ice core | (Mackay et al., 2022) |
| Ilopango (TBJ) | 450 (±30) CE | 431 (±2) CE | Ice core | (Smith et al., 2020) |
| Okmok II | 76 BCE | 43 BCE | Ice core, dendroclimatology | (McConnell et al., 2020) |
| Santorini | 1610 (±14) BCE | 1609–1560 BCE (95.4% probability) | Radiocarbon, dendrochronology | (Manning, 2022) |
| Aniakchak II | 1645 (±10) BCE | 1628 (±1) BCE | Ice core, dendroclimatology | (Pearson et al., 2022) |
| Laacher See | 12,916 BP | 13,006 (±9) BP | Radiocarbon, dendrochronology | (Reinig et al., 2021) |
| Taupō (Oruanui) | 27,100 (±960) BP | 25,319 (±250) BP | Ice core | (Dunbar et al., 2017; Sigl et al., 2016) |
| | | 25,335–25,800 BP (95.4% probability) | Radiocarbon | (Muscheler et al., 2020) |

I invite the authors of this paper to address my comments. I suggest that these amendments could be made in the corresponding tables of this manuscript, providing a basis for updating the PalVol reconstruction in its next iteration. You could also mention in this manuscript that there is a floating continuous volcanic forcing record for the time period 12.8-13.2 ka BP, aimed to support transient model simulations for the Younger Dryas inception (Abbott et al., 2021).

**REFERENCES:**

Abbott, P. M., Niemeier, U., Timmreck, C., Riede, F., McConnell, J. R., Severi, M., Fischer, H., Svensson, A., Toohey, M., Reinig, F., and Sigl, M.: Volcanic climate forcing preceding the inception of the Younger Dryas: Implications for tracing the Laacher See eruption, *Quaternary Sci Rev*, 274, 2021.

Crosweller, H. S., Arora, B., Brown, S. K., Cottrell, E., Deligne, N. I., Guerrero, N. O., Hobbs, L., Kiyosugi, K., Loughlin, S. C., Lowndes, J., Nayembil, M., Siebert, L., Sparks, R. S. J., Takarada, S., and Venzke, E.: Global database on large magnitude explosive volcanic eruptions (LaMEVE), *Journal of Applied Volcanology*, 1, 4, 2012.

Dunbar, N. W., Iverson, N. A., Van Eaton, A. R., Sigl, M., Alloway, B. V., Kurbatov, A. V., Mastin, L. G., McConnell, J. R., and Wilson, C. J. N.: New Zealand supereruption provides time marker for the Last Glacial Maximum in Antarctica, *Sci Rep-Uk*, 7, 2017.

Lin, J., Svensson, A., Hvidberg, C. S., Lohmann, J., Kristiansen, S., Dahl-Jensen, D., Steffensen, J. P., Rasmussen, S. O., Cook, E., Kjær, H. A., Vinther, B. M., Fischer, H., Stocker, T., Sigl, M., Bigler, M., Severi, M., Traversi, R., and Mulvaney, R.: Magnitude, frequency and climate forcing of global volcanism during the last glacial period as seen in Greenland and Antarctic ice cores (60–9ka BP), *Clim. Past*, 18, 485-506, 2022.

Mackay, H., Plunkett, G., Jensen, B. J. L., Aubry, T. J., Corona, C., Kim, W. M., Toohey, M., Sigl, M., Stoffel, M., Anchukaitis, K. J., Raible, C., Bolton, M. S. M., Manning, J. G., Newfield, T. P., Di Cosmo, N., Ludlow, F., Kostick, C., Yang, Z., Coyle McClung, L., Amesbury, M., Monteath, A., Hughes, P. D. M., Langdon, P. G., Charman, D., Booth, R., Davies, K. L., Blundell, A., and Swindles, G. T.: The 852/3 CE Mount Churchill eruption: examining the potential climatic and societal impacts and the timing of the Medieval Climate Anomaly in the North Atlantic region, *Clim. Past*, 18, 1475-1508, 2022.

Manning, S. W.: Second Intermediate Period date for the Thera (Santorini) eruption and historical implications, *Plos One*, 17, 2022.

McConnell, J. R., Sigl, M., Plunkett, G., Burke, A., Kim, W. M., Raible, C. C., Wilson, A. I., Manning, J. G., Ludlow, F., Chellman, N. J., Innes, H. M., Yang, Z., Larsen, J. F., Schaefer, J. R., Kipfstuhl, S., Mojtabavi, S., Wilhelms, F., Opel, T., Meyer, H., and Steffensen, J. P.: Extreme climate after massive eruption of Alaska's Okmok volcano in 43 BCE and effects on the late Roman Republic and Ptolemaic Kingdom, *P Natl Acad Sci USA*, 117, 15443-15449, 2020.

Muscheler, R., Adolphi, F., Heaton, T. J., Bronk Ramsey, C., Svensson, A., van der Plicht, J., and Reimer, P. J.: Testing and Improving the IntCal20 Calibration Curve with Independent Records, *Radiocarbon*, 62, 1079-1094, 2020.

Oppenheimer, C., Orchard, A., Stoffel, M., Newfield, T. P., Guillet, S., Corona, C., Sigl, M., Di Cosmo, N., and Buntgen, U.: The Eldgja eruption: timing, long-range impacts and influence on the Christianisation of Iceland, *Climatic Change*, 147, 369-381, 2018.

Oppenheimer, C., Wacker, L., Xu, J., Galvan, J. D., Stoffel, M., Guillet, S., Corona, C., Sigl, M., Di Cosmo, N., Hajdas, I., Pan, B., Breuker, R., Schneider, L., Esper, J., Fei, J., Hammond, J. O. S., and Büntgen, U.: Multi-proxy dating the 'Millennium Eruption' of Changbaishan to late 946 CE, *Quaternary Sci Rev*, 158, 164-171, 2017.

Pearson, C., Sigl, M., Burke, A., Davies, S., Kurbatov, A., Severi, M., Cole-Dai, J., Innes, H., Albert, P. G., and Helmick, M.: Geochemical ice-core constraints on the timing and climatic impact of Aniakchak II (1628 BCE) and Thera (Minoan) volcanic eruptions, *PNAS Nexus*, doi: 10.1093/pnasnexus/pgac048, 2022.

Plunkett, G., Sigl, M., McConnell, J. R., Pilcher, J. R., and Chellman, N. J.: The significance of volcanic ash in Greenland ice cores during the Common Era, *Quaternary Sci Rev*, 301, 107936, 2023.

Reinig, F., Wacker, L., Jöris, O., Oppenheimer, C., Guidobaldi, G., Nievergelt, D., Adolphi, F., Cherubini, P., Engels, S., Esper, J., Land, A., Lane, C., Pfanz, H., Remmele, S., Sigl, M., Sookdeo, A., and Büntgen, U.: Precise date for the Laacher See eruption synchronizes the Younger Dryas, *Nature*, 595, 66-69, 2021.

Sigl, M., Fudge, T. J., Winstrup, M., Cole-Dai, J., Ferris, D., McConnell, J. R., Taylor, K. C., Welten, K. C., Woodruff, T. E., Adolphi, F., Bisiaux, M., Brook, E. J., Buizert, C., Caffee, M. W., Dunbar, N. W., Edwards, R., Geng, L., Iverson, N., Koffman, B., Layman, L., Maselli, O. J., McGwire, K., Muscheler, R., Nishiizumi, K., Pasteris, D. R., Rhodes, R. H., and Sowers, T. A.: The WAIS Divide deep ice core WD2014 chronology - Part 2: Annual-layer counting (0-31 ka BP), *Clim Past*, 12, 769-786, 2016.

Sigl, M., Toohey, M., McConnell, J. R., Cole-Dai, J., and Severi, M.: Volcanic stratospheric sulfur injections and aerosol optical depth during the Holocene (past 11 500 years) from a bipolar ice-core array, *Earth System Science Data*, 14, 3167-3196, 2022.

Sigl, M., Winstrup, M., McConnell, J. R., Welten, K. C., Plunkett, G., Ludlow, F., Büntgen, U., Caffee, M., Chellman, N., Dahl-Jensen, D., Fischer, H., Kipfstuhl, S., Kostick, C., Maselli, O. J., Mekhaldi, F., Mulvaney, R., Muscheler, R., Pasteris, D. R., Pilcher, J. R., Salzer, M., Schupbach, S., Steffensen, J. P., Vinther, B. M., and Woodruff, T. E.: Timing and climate forcing of volcanic eruptions for the past 2,500 years, *Nature*, 523, 543-549, 2015.

Smith, V. C., Costa, A., Aguirre-Diaz, G., Pedrazzi, D., Scifo, A., Plunkett, G., Poret, M., Tournigand, P. Y., Miles, D., Dee, M. W., McConnell, J. R., Sunye-Puchol, I., Harris, P. D., Sigl, M., Pilcher, J. R., Chellman, N., and Gutierrez, E.: The magnitude and impact of the 431 CE Tierra Blanca Joven eruption of Ilopango, El Salvador, *P Natl Acad Sci USA*, 117, 26061-26068, 2020.

Toohey, M. and Sigl, M.: Volcanic stratospheric sulfur injections and aerosol optical depth from 500 BCE to 1900 CE, *Earth System Science Data*, 9, 809-831, 2017.

---

## Author Response (AR1)

**RC1**: 'Comment on essd-2023-103', Anonymous Referee #1, 30 Jul 2023

This paper aims to extend the proxy of volcanic stratospheric sulfur injections over the last 130,000 years by combining the tephra volcanic records and ice-core volcanic records and simulating the volcanic eruption rate and magnitude. The idea is fresh, and it is a good try to extend the volcanic record to the glacial period. I think this paper needs a thorough consideration of several points in the following.

The underlying basis for this reconstructed proxy is the linear relationship between the erupted tephra volume and the sulfur gas volume. However, explosive volcanoes are not always sulfur-rich volcanoes. So, the methodology of using the LaMEVE dataset to simulate the distribution of sulfur-rich eruption sites is not precise.

*Thanks for this comment. We completely agree that the calculation of the sulfate from the magnitude is not precise for individual eruptions, and we do not want users to expect that the estimates are precise. We state this explicitly in the manuscript, e.g.:*

- o *Line 25; "While the reconstruction often differs from ice core estimates for specific eruptions due to uncertainties in the data used and reconstruction method…"*
- o *Line 309: "Accordingly, VSSI estimated from eruption magnitudes should be understood to have significant uncertainty for any individual eruption…"*
- o *Line 479: "Importantly, the VSSI values for individual eruptions have significant uncertainties, as we (and prior works; e.g., Andres et al., 1993; Sigurdsson, 1990) showed that the amount of sulfur released by an eruption can vary by orders of magnitude for any given eruption magnitude…"*

*Nonetheless, as we state at line 488: "although the VSSI for any specific eruption is quite uncertain, we expect that due to the observed relationship between VSSI and magnitude, this will be compensated when averaged over a sufficiently large set of eruptions, and therefore the tephra data may contain information on variations in VSSI on long time scales". In other words, we believe that there is value in the reconstruction even if the estimated VSSI for each individual eruption is quite uncertain.*

The volcanic simulation is based on the constant eruption rate in the Holocene, which is not the case in the last glacial period, which has more large sulfur-rich volcanic eruptions as seen from ice-core volcanic records. The erupted situation in the Holocene is quite different from that in the last glacial period.

*We appreciate that it is very difficult to estimate the eruption frequency during the last glacial period. In terms of synchronized bipolar ice core records—the gold standard for estimating eruption in the past—we are aware only of the 2022 study by Lin et al., who we cite at line 85. From their conclusions:*

*"Overall, the frequency of volcanic eruptions per millennium is rather constant throughout the investigated period and comparable to that of the most recent millennia. In agreement with previous studies, however, we find elevated levels of volcanic activity in the NH during the deglacial period (16–9 kab2k)".*

*This is the basis of our assumed constant "baseline" eruption frequency. As shown in Sec. 3.3 of our manuscript, our method is able to reproduce the elevated rate of large magnitude eruptions during the deglacial period directly from the tephra data.*

The author should be more careful when using the stochastic approach to revise small-magnitude eruptions in the last glacial period.

*We do not revise or change the estimated magnitudes of eruptions in the LaMEVE database. The stochastic method only fills the incomplete tephra record with hypothetical eruptions with statics based on the Holocene ice core record.*

In this PalVol proxy, the uncertainty of VSSI for this semi-stochastic ensemble reconstruction proxy is not estimated, and the magnitude of VSSI and the timing of volcanoes are constrained in the millennium range, which may limit its further use to estimate abrupt climate change.

*Our provided forcing data does in fact include uncertainties in the estimated VSSI derived from tephras: we have put a greater emphasis on this fact in the revised manuscript. It is true the timing of eruptions has large uncertainties for some eruptions which does limit the utility of this data in comparison to climate proxies, but we would argue, does not limit the utility of the forcing in idealized model studies of abrupt climate change.*

**Detailed comments:**

Abstract:

Line 19  This sentence is not precise. The continuous record of stratospheric sulfur injections and aerosol optical proxy have only been reconstructed in the Holocene.

*The work of Lin et al. (2022) provides estimates of VSSI extending back to 60,000 BP. It is true that together there is not a continuous record that stretches over this full period, but we do not claim that there is, only that "information from ice cores has been used to derive estimates of stratospheric sulfur injections…"*

Introduction:

Please add references in paragraph 2.

*We have added some references as requested although much of this is textbook knowledge.*

Line 85 This is wrong. The frequency of large eruptions in the last glacial period is higher compared that in the Holocene period.

*We are unaware of what the referee is basing this statement on. We are basing our assumption on the recent findings of Lin et al., 2022 as discussed above.*

Data and method:

Deriving VSSI from tephra: the assumption is the linear relationship between VSSI and erupted volume. The theory behind this assumption is that all volcanic eruptions are sulfur-rich eruptions, which is not the real case and could over-estimate the VSSI.

*The linear fit is consistent with the data shown in Fig. 4, which shows that for any given magnitude (i.e., erupted volume) there is a range of VSSI amounts supported by satellite observations and ice core derived values. The least squares fit finds the line that minimizes the differences between the fit and the data points, i.e., the fit goes through the center of the data. Based on the data, the fit will overestimate the VSSI for some eruptions but underestimate it for others. We do not believe that this method assumes that all eruptions are sulfur-rich—we attempt to derive a relationship which applies to all eruptions on average.*

Line 234 Please add specific date period.

*This sentence has been edited to give the specific date period.*

Section 2.4  It's necessary to show the assumed volcanic distribution that is used to simulate the timing and magnitude for individual volcanic eruptions. To my knowledge, the last glacial period has relative more large eruptions compared to that in the Holocene period. If so, please use the revised volcanic distribution for the last glacial period as the input to generate the synthetic volcanic time series.

*We are unaware of the data on which the reviewer is basing the statement that the last glacial period was more volcanically active than the Holocene. Our method is based here on the results of Lin et al. 2022 who state that the baseline eruption frequency during the glacial period is not significantly different than recent millennia (see above).*

Results:

Line 291: Large eruption magnitudes generally lead to larger VSSI values. I would say this is not precise.

*We do not claim that the method is precise for individual eruptions, but that it should be unbiased over large samples of eruptions. That is our goal.*

Line 292: This fitted power law formula is not precise. The data of comparison between VSSI and volcanic magnitude is collected from different approaches. One is from observed satellite and the other is from measured ice core.  The uncertainty from different approaches is not estimated in this paper. This comparison is not convincing.

*The satellite and ice core estimates of VSSI certainly both have uncertainties, which can be quite significant. Nonetheless, the fact that the two methods show reasonable overlap in their relationship with eruption magnitude is, we find, an encouraging result, and the power-law relationship appears to describe the increase in VSSI with eruption volume suggested by both data sets. Again, this power-law relationship will not be precise for any single eruption, but is constructed to be unbiased when many eruptions are averaged together.*

*Uncertainties in VSSI for each individual eruption are included in the data file. This was stated in the conclusions "Uncertainties in VSSI are included in our reconstruction as upper and lower bounds on the VSSI for each eruption, based on the uncertainty in our derived magnitude to VSSI relationship." We have added a similar comment in the Method section 2.3.*

**RC2**: ['Comment on essd-2023-103'](), Anonymous Referee #2, 01 Aug 2023

The paper of Schindlbeck-Belo et al. proposes a new reconstruction of the stratospheric sulfur injection of volcanic origin for the last 130 000 years with the objective to improve the modeling of the climate variations over millennial time scales by better taking into account volcanic forcing.

The paper is globally well written and particularly well introduced. I think the paper may be a good contribution to ESSD but first I have some points I would like to rise.

The period considered, e.g. 130 kyrs, is featured by glacial/interglacial variations. The authors indicate in the introduction (lines 75-83) and later in the manuscript (ex: lines 360, conclusion) that both ice core sulfur and tephra records attest a "marked increase in eruption frequencies during and after the last glaciation, especially in the northern hemisphere (NH) mid-to-high latitudes" (lines 76-77).

A possible variability of the climate on these time scales could therefore be expected – depending of course, on the magnitude of the eruptions.

Why did the authors consider a static distribution of eruption frequency, distribution they also qualified of true (line 244)? This distribution was considered not only for the testing period (12 to 0 kyrs) but also for building the final time-series. This does not seem to be the most representative distribution of eruptions over such long time scales, taking into account the observations (HolVol but not only). It therefore introduces an artificial variability into the new time-series

*We did consider baking a variable eruption frequency into the reconstruction method, and indeed this might be something to explore in future research. However, for the current purpose of a reconstruction for the last 140 ka, we found here that it was unnecessary: even when assuming a constant baseline eruption frequency distribution, the use of the tephra data produced an increase in cumulative VSSI in the early Holocene in line with the ice core data (see Sec 3.3 and Fig 7). We find this a novel illustration of the power of the tephra data even given its incompleteness.*

Actually, tephra and ice cores data provide complementary information which may or not overlap. Merging the two data sets is a good idea. However, it seems that the authors have given priority to the information provided by tephra by integrating it to the synthetic data (lines 241-246). This may participates to create a bias as the one of Fig 7a. There, the number of events per millennia as obtained for PalVol is relatively stable, between 12 and 2 kyrs, but is increased by 2 between 2 kyrs to present, similarly as tephra. This bias may also result from the over-counting of tephra events as suggested by the authors (lines 360-365), but likely not alone.

*Yes, our aim is to utilize the tephra data as much as possible, as it has the advantage of extending further into the past than ice cores. It is true that the tephra record includes many more events in the last millennium than do the ice cores, illustrated by the bias in Fig. 7a referenced by the reviewer. This does not, however, seem to have a large impact on the cumulative VSSI: one can see in Fig 7b a good agreement between the ice core and tephra VSSI amount. One can also see in Fig. 7c that the VSSI per event is*

*significantly smaller for tephra than for ice core in the last millennium—suggesting that many of the eruptions included in the tephra records for the past millennium are of the M=4 category that contribute only a small amount to the cumulative VSSI compared to larger eruptions.*

*There may certainly be other aspects to this bias, but our aim is primarily to produce a reconstruction that extends beyond that currently possible with ice cores. The most recent millennia are included here for completeness and for validating our method against the ice core records. For applications, though, we quite expect (and encourage) the use of ice core-based reconstructions for the most recent millennia, so we are not overly concerned with biases that concern only the most recent millennia.*

No information is provided on the statistics of the data sets considered to build the new time-series. Several references are made to it (lines 229, 241, 353).

*We do not explicitly diagnose the statistics of the HolVol record used as a basis for our synthetic event timeseries. Characterization of the LaMEVE data was performed by prior studies, e.g., Brown et al. (2014). The resampling method we use (that of Bethke et al., 2017) will reproduce the "statistics" of the input data, e.g., the eruption magnitude-frequency distribution, simply by resampling (or "shuffling") the input data. Our references to the "statistics" are not specific to any calculated descriptive statistics, but refer to the general statistical properties of the input and output data. We have edited the text in the 3 locations referred to, replacing "statistics" with "statistical characteristics".*

Actually, a large number of small events per millennia may be equivalent to a very large eruption, in terms of VSSI. How do the authors deal with this aspect?

*The following table lists the number of eruptions in the tephra database back to 130,000 BP for magnitudes 4 to 9 along with the total VSSI associated with eruptions of each magnitude:*

| Magnitude | Number of events | Total VSSI (TgS) |
|---|---|---|
| 4 | 797 | 432 |
| 5 | 505 | 1650 |
| 6 | 194 | 3995 |
| 7 | 38 | 4197 |
| 8 | 2 | 952 |
| 9 | 1 | 2984 |

*Based on the tephra data (with its sampling incompleteness), the most important eruption magnitudes in term of contribution to the total stratospheric aerosol forcing are magnitudes 6 and 7 (with the combined M>=8 contributing a similar amount). Of course, magnitudes 4 and 5 are not insignificant in reality, and this is what has motivated our use of the synthetic eruption timeseries, to fill the data set with hypothetical events and improve the statistical aspects of the forcing.*

Minor comments:

In the introduction, a good comparison is made between ice core and marine sediments. Could you possibly add information on which one provides better constrains on volume estimates and so more realistic magnitude?

*Ice core volcanic signals do not provide any direct information about the volume or magnitude, since they only provide information on the sulfur output of eruptions.*

Line 44: sulfate is not electronically neutral, please replace SO4 by $SO_4^{2-}$

line 51 missing reference in the bibliography: *Robock 2000*

line 58: Is there any missing word after Glacial? *– added "period"*

line 106: VSSI acronym already introduced page 1

*According to the Copernicus style guide, acronyms "need to be defined in the abstract and then again at the first instance in the rest of the text".*

line 122-23. "with the frequency of smaller eruptions (M=4) falling off much faster than that for larger eruptions (M>6)" → can you be more precise? It seems that there are intermediate magnitudes between 4 & 5 for instance, that seem to be more under-reported than M=4.

*Thanks for pointing this out. What we meant are eruptions between 4.0<=M<5.0. We changed this accordingly.*

Figure 2: I don't know if this is possible, but it may be interesting to see this plot with two colours, one showing data from marine sediments, one from ice-cores allowing to keep in mind the limits of estimates and somehow their uncertainties which are difficult to estimate

*This is a compilation from the LaMEVE database and additional marine tephra data. Most of the ice core age data is from the last 10,000 years=most eruptions in the record, you will not see much. Marine data more equally distributed.*

line 132: missing reference: *added*

line 133: last glacial cycle → give the corresponding time interval: *done*

line 134: "small to medium eruptions" → give the range of VEI: *done*

line 148-149: check the second part of the sentence. There should be a VEI >= 4. *done*

It is unclear if the database is only based on tephra compiled from continental environments.: *the database encompasses all known eruptions from all geological settings.*

line 173 : check bracket sequence: *seems correct*

line 194: how many eruptions for this period? *Added number*

line 206: Uncertainties in the VSSI values in HolVol are typically around 35%. Can you say more or indicate a reference?

*This sentence modified to: Reported uncertainties in the VSSI values in HolVol for explosive (i.e., non-effusive) eruptions are typically between approximately 20 and 40% (Sigl et al., 2021).*

line 229: "with the same statistics as an input data set" Can you be more precise? What do you mean by same statistics?

*We have modified this sentence to use the term "statistical characteristics" by which we mean things like the long term average, the return time for a given VSSI value. To not over complicate this sentence we have not included examples, but trust that the small edit along with the discussion in the text of centennial and millennial cumulative VSSI amounts will make it clear.*

Lines 235-239: the description of the generation of random eruption parameters is confusing: eruptions with random parameters that not identified in the data base.

*Thank you for pointing this out. We have adjusted the sentence to:*

*"For eruptions that are unidentified in the HolVol base data (which is the majority of events), eruption parameters month and precise latitude (within three latitude ranges) are unknown, and set to default values. In our synthetic eruptions, the eruption month is randomized uniformly across the calendar year, and the eruption latitude is randomized within the identified tropical, NH and SH bands using the probability density of the LaMEVE data set between 10 ka BP and the present."*

Line 258: do you have references for the magnitude estimates of these 2 eruptions?

*References added*

Fig 7c: Why not using the median instead of the mean? The mean estimate is more biaised by outliers.

*This is true and the median might be a useful descriptor of the data. Here though we are aiming to understand the timeseries in panel b. Panel c shows the average VSSI per event, which is numerically equal to the values of panel b divided by the number of events per millennium in panel a. For a median, this simple relation to the quantities shown in panels a and b would not hold.*

Fig 7d. Can you explain more this correlation histogram? What does represent the fraction?

*In the histogram, the height of each bar is the relative number or "fraction" of observations (number of ensemble members in bin / total number of ensemble members), and the sum of the bar heights is less than or equal to 1.*

*We have edited the relevant portion of the figure caption to: (d) histogram of correlation coefficients calculated between the ensemble of semi-synthetic millennial cumulative VSSI time series with the HolVol time series. Each bar indicates the fraction of all ensemble members with correlation coefficients between the values defining the edges of the bar along the horizontal axis.*

Fig 8: Check Reference: Lin et al 2021 (in text) or 2022? *- checked*

**Community Comment: Michael Sigl**

I would like to add a short comment to this paper. In the absence of a continuous dataset of stratospheric volcanic sulfur emission and aerosol optical properties datasets prior to 11,500 years BP, I think it is a good idea to get comparable information from a synthetic product that takes into account ice core data but also tephra data from proximal deposits.

I also acknowledge, that you are emphasizing inherent limitations of this synthetic product (e.g. comparably large error bars for age and strength of VSSI), and suggest to use or combine with, where possible, more precise data derived from ice-core records (Lin et al., 2022; Sigl et al., 2022; Toohey and Sigl, 2017). However, user of the dataset will likely employ it for many purposes, including linking the eruption record to rapid climate change, extreme events and societal collapse. A lot of progress has been achieved in the past decade regarding the dating of past eruptions using for example new-generation ice-core analyses or annual-resolution radiocarbon analyses. The LaMEVE database which forms the backbone of the PalVol v1 reconstruction, however, was built over a decade ago (Crosweller et al., 2012), and represents an earlier state of research, which has since been modified or refined many times.

*Thanks a lot for your comment. You are absolutely right that the version of the LaMEVE database that we used has not been updated for the latest age datings. However, the database is frequently updated and we used the latest version (Version 3), furthermore we also updated several ages by our own and aim to include the new ages in the next version of PalVol.*

*We added this paragraph in the text: "The current literature has updated several eruption ages, which have not been included in PalVol v1, but which will be included in the next version. These comprise especially ages obtained by ice core or dendrochronological studies (e.g., Bárdarbunga 877 CE (Plunkett et al., 2023); White River Ash 853 CE (Mackay et al., 2022); Ilopango 431 CE (Smith et al., 2020); Okmok II 43 BCE (McConnell et al., 2020); Aniakchak II 1628 BCE (Pearson et al., 2022); Laacher See 13,006 BP (Reinig et al., 2021))."*

Upon noting numerous eruption ages in Table A1 which differ from what is currently considered the best estimates, I have downloaded the LaMEVE eruption catalogue (https://www2.bgs.ac.uk/vogripa/view/controller.cfc?method=lameve) and critically assessed the dating of past eruptions in this dataset against revised age estimates published since. My analysis is incomplete and biased towards Late Glacial, Holocene and Common Era eruptions. Dating errors are considered to represent $2\sigma$ uncertainties unless stated otherwise.

I invite the authors of this paper to address my comments. I suggest that these amendments could be made in the corresponding tables of this manuscript, providing a basis for updating the PalVol reconstruction in its next iteration.

You could also mention in this manuscript that there is a floating continuous volcanic forcing record for the time period 12.8-13.2 ka BP, aimed to support transient model simulations for the Younger Dryas inception (Abbott et al., 2021).

*Thanks for this, we added the reference and a short sentence to our introduction.*

| Volcano | Date (PALVOL) | Date (revised) | Method for revision | References |
|---|---|---|---|---|
| Changbaishan | 942 (±4) CE | Nov 946 CE | Dendrochronology, ice core, radiocarbon | (Oppenheimer et al., 2017) |
| Katla (Eldgjá) | 934 (±2) CE | 939 CE | Documentary, ice core | (Oppenheimer et al., 2018; Sigl et al., 2015) |
| Bárdarbunga (Vatnaoldur) | 870 CE | 877 (±1) CE | Ice core | (Plunkett et al., 2023) |
| Churchill (White River Ash, eastern lobe (WRAe) | 847 (±1) CE | 853 (±1) CE | Ice core | (Mackay et al., 2022) |
| Ilopango (TBJ) | 450 (±30) CE | 431 (±2) CE | Ice core | (Smith et al., 2020) |
| Okmok II | 76 BCE | 43 BCE | Ice core, dendroclimatology | (McConnell et al., 2020) |
| Santorini | 1610 (±14) BCE | 1609–1560 BCE (95.4% probability) | Radiocarbon, dendrochronology | (Manning, 2022) |
| Aniakchak II | 1645 (±10) BCE | 1628 (±1) BCE | Ice core, dendroclimatology | (Pearson et al., 2022) |
| Laacher See | 12,916 BP | 13,006 (±9) BP | Radiocarbon, dendrochronology | (Reinig et al., 2021) |
| Taupō (Oruanui) | 27,100 (±960) BP | 25,319 (±250) BP | Ice core | (Dunbar et al., 2017; Sigl et al., 2016) |
|  |  | 25,335–25,800 BP (95.4% probability) | Radiocarbon | (Muscheler et al., 2020) |

REFERENCES:

Abbott, P. M., Niemeier, U., Timmreck, C., Riede, F., McConnell, J. R., Severi, M., Fischer, H., Svensson, A., Toohey, M., Reinig, F., and Sigl, M.: Volcanic climate forcing preceding the inception of the Younger Dryas: Implications for tracing the Laacher See eruption, Quaternary Sci Rev, 274, 2021.

Crosweller, H. S., Arora, B., Brown, S. K., Cottrell, E., Deligne, N. I., Guerrero, N. O., Hobbs, L., Kiyosugi, K., Loughlin, S. C., Lowndes, J., Nayembil, M., Siebert, L., Sparks, R. S. J., Takarada, S., and Venzke, E.: Global database on large magnitude explosive volcanic eruptions (LaMEVE), Journal of Applied Volcanology, 1, 4, 2012.

Dunbar, N. W., Iverson, N. A., Van Eaton, A. R., Sigl, M., Alloway, B. V., Kurbatov, A. V., Mastin, L. G., McConnell, J. R., and Wilson, C. J. N.: New Zealand supereruption provides time marker for the Last Glacial Maximum in Antarctica, Sci Rep-Uk, 7, 2017.

Lin, J., Svensson, A., Hvidberg, C. S., Lohmann, J., Kristiansen, S., Dahl-Jensen, D., Steffensen, J. P., Rasmussen, S. O., Cook, E., Kjær, H. A., Vinther, B. M., Fischer, H., Stocker, T., Sigl, M., Bigler, M., Severi, M., Traversi, R., and Mulvaney, R.: Magnitude, frequency and climate forcing of global volcanism during the last glacial period as seen in Greenland and Antarctic ice cores (60–9ka BP), Clim. Past, 18, 485-506, 2022.

Mackay, H., Plunkett, G., Jensen, B. J. L., Aubry, T. J., Corona, C., Kim, W. M., Toohey, M., Sigl, M., Stoffel, M., Anchukaitis, K. J., Raible, C., Bolton, M. S. M., Manning, J. G., Newfield, T. P., Di Cosmo, N., Ludlow, F., Kostick, C., Yang, Z., Coyle McClung, L., Amesbury, M., Monteath, A., Hughes, P. D. M., Langdon, P. G., Charman, D., Booth, R., Davies, K. L., Blundell, A., and Swindles, G. T.: The 852/3 CE Mount Churchill eruption: examining the potential climatic and societal impacts and the timing of the Medieval Climate Anomaly in the North Atlantic region, Clim. Past, 18, 1475-1508, 2022.

Manning, S. W.: Second Intermediate Period date for the Thera (Santorini) eruption and historical implications, Plos One, 17, 2022.

McConnell, J. R., Sigl, M., Plunkett, G., Burke, A., Kim, W. M., Raible, C. C., Wilson, A. I., Manning, J. G., Ludlow, F., Chellman, N. J., Innes, H. M., Yang, Z., Larsen, J. F., Schaefer, J. R., Kipfstuhl, S., Mojtabavi, S., Wilhelms, F., Opel, T., Meyer, H., and Steffensen, J. P.: Extreme climate after massive eruption of Alaska's Okmok volcano in 43 BCE and effects on the late Roman Republic and Ptolemaic Kingdom, P Natl Acad Sci USA, 117, 15443-15449, 2020.

Muscheler, R., Adolphi, F., Heaton, T. J., Bronk Ramsey, C., Svensson, A., van der Plicht, J., and Reimer, P. J.: Testing and Improving the IntCal20 Calibration Curve with Independent Records, Radiocarbon, 62, 1079-1094, 2020.

Oppenheimer, C., Orchard, A., Stoffel, M., Newfield, T. P., Guillet, S., Corona, C., Sigl, M., Di Cosmo, N., and Buntgen, U.: The Eldgja eruption: timing, long-range impacts and influence on the Christianisation of Iceland, Climatic Change, 147, 369-381, 2018.

Oppenheimer, C., Wacker, L., Xu, J., Galvan, J. D., Stoffel, M., Guillet, S., Corona, C., Sigl, M., Di Cosmo, N., Hajdas, I., Pan, B., Breuker, R., Schneider, L., Esper, J., Fei, J., Hammond, J. O. S., and Büntgen, U.: Multi-proxy dating the 'Millennium Eruption' of Changbaishan to late 946 CE, Quaternary Sci Rev, 158, 164-171, 2017.

Pearson, C., Sigl, M., Burke, A., Davies, S., Kurbatov, A., Severi, M., Cole-Dai, J., Innes, H., Albert, P. G., and Helmick, M.: Geochemical ice-core constraints on the timing and climatic impact of Aniakchak II (1628 BCE) and Thera (Minoan) volcanic eruptions, PNAS Nexus, doi: 10.1093/pnasnexus/pgac048, 2022.

Plunkett, G., Sigl, M., McConnell, J. R., Pilcher, J. R., and Chellman, N. J.: The significance of volcanic ash in Greenland ice cores during the Common Era, Quaternary Sci Rev, 301, 107936, 2023.

Reinig, F., Wacker, L., Jöris, O., Oppenheimer, C., Guidobaldi, G., Nievergelt, D., Adolphi, F., Cherubini, P., Engels, S., Esper, J., Land, A., Lane, C., Pfanz, H., Remmele, S., Sigl, M., Sookdeo, A., and Büntgen, U.: Precise date for the Laacher See eruption synchronizes the Younger Dryas, Nature, 595, 66- 69, 2021.

Sigl, M., Fudge, T. J., Winstrup, M., Cole-Dai, J., Ferris, D., McConnell, J. R., Taylor, K. C., Welten, K. C., Woodruff, T. E., Adolphi, F., Bisiaux, M., Brook, E. J., Buizert, C., Caffee, M. W., Dunbar, N. W., Edwards, R., Geng, L., Iverson, N., Koffman, B., Layman, L., Maselli, O. J., McGwire, K., Muscheler, R., Nishiizumi, K., Pasteris, D. R., Rhodes, R. H., and Sowers, T. A.: The WAIS Divide deep ice core WD2014 chronology - Part 2: Annual-layer counting (0-31 ka BP), Clim Past, 12, 769-786, 2016.

Sigl, M., Toohey, M., McConnell, J. R., Cole-Dai, J., and Severi, M.: Volcanic stratospheric sulfur injections and aerosol optical depth during the Holocene (past 11 500 years) from a bipolar ice-core array, Earth System Science Data, 14, 3167-3196, 2022.

Sigl, M., Winstrup, M., McConnell, J. R., Welten, K. C., Plunkett, G., Ludlow, F., Büntgen, U., Caffee, M., Chellman, N., Dahl-Jensen, D., Fischer, H., Kipfstuhl, S., Kostick, C., Maselli, O. J., Mekhaldi, F., Mulvaney, R., Muscheler, R., Pasteris, D. R., Pilcher, J. R., Salzer, M., Schupbach, S., Steffensen, J. P., Vinther, B. M., and Woodruff, T. E.: Timing and climate forcing of volcanic eruptions for the past 2,500 years, Nature, 523, 543-549, 2015.

Smith, V. C., Costa, A., Aguirre-Diaz, G., Pedrazzi, D., Scifo, A., Plunkett, G., Poret, M., Tournigand, P. Y., Miles, D., Dee, M. W., McConnell, J. R., Sunye-Puchol, I., Harris, P. D., Sigl, M., Pilcher, J. R., Chellman, N., and Gutierrez, E.: The magnitude and impact of the 431 CE Tierra Blanca Joven eruption of Ilopango, El Salvador, P Natl Acad Sci USA, 117, 26061-26068, 2020.

Toohey, M. and Sigl, M.: Volcanic stratospheric sulfur injections and aerosol optical depth from 500 BCE to 1900 CE, Earth System Science Data, 9, 809-831, 2017.